# The adaptive growth and mechanisms of *Klebsiella pneumoniae* under sucrose and glucose exposure

Yunhui He,[1,2] Fangfang Liu,[2] Congcong Li,[1,2] Jiayan Wu,[2] Kewei Fan,[2] Zewen Wen,[2] Duoyun Li,[2] Zhijian Yu,[1,2,3] Tieying Hou[2]

**ABSTRACT** *Klebsiella pneumoniae* commonly colonizes the gastrointestinal (GI) mucosa of animals and healthy humans. Successful GI colonization by *K. pneumoniae* requires overcoming the colonization resistance (CR) exerted by the gut microbiota. Although previous studies have demonstrated the role of microbial carbohydrate metabolism in *K. pneumoniae* colonization, the specific effects of individual carbohydrates, such as glucose and sucrose, particularly across concentration gradients or under sustained induction on the adaptive growth of *K. pneumoniae* remain unknown. Herein, we demonstrate that 4% or 8% glucose and sucrose are favorable for promoting adaptive growth of sugar-induced strains. Additionally, the growth response to glucose exhibited strain-specific variability. Sustained sugar induction did not significantly alter the hypermucoviscosity (HMV) phenotype but did affect antibiotic resistance of *K. pneumoniae*. Knockout of the *scrA* and *scrY* genes impaired the adaptive growth under sucrose and glucose conditions, yet did not significantly influence antimicrobial susceptibility or the HMV phenotype. These findings provide insights into the metabolic regulation of *K. pneumoniae* colonization and offer potential guidance for clinical treatment strategies targeting *K. pneumoniae*-associated infections.

**IMPORTANCE** This work elucidates the impact of single-carbon source gradients and sustained sugar induction on the adaptive growth and drug resistance of *Klebsiella pneumoniae* and preliminarily reveals the roles of *scrA* and *scrY* in carbohydrate metabolism, suggesting a possible mechanism by which sucrose and glucose affect the adaptive growth of *K. pneumoniae*. These findings contribute to the theoretical understanding of CCR and provide insights that may inform clinical management of *K. pneumoniae*-related infections.

**KEYWORDS** *Klebsiella pneumoniae*, sucrose, glucose, adaptive growth, *scrA*, scrY

K*lebsiella pneumoniae*, a member of the *Enterobacteriaceae* family, can be classified into classical *K. pneumoniae* (cKP) and hypervirulent *K. pneumoniae* (hvKP) based on differences in virulence and pathogenicity. HvKp harbors conserved virulence gene clusters, including *rmpA/rmpA*[2] (regulators of the hypermucoviscous phenotype), *iroBCD* (involved in iron acquisition), and *iucABCD* (responsible for siderophore synthesis). Importantly, several of these virulence genes are plasmid-borne, enabling horizontal transfer and the potential acquisition of hypervirulent traits by otherwise classical strains (1). CKP primarily affects immunocompromised individuals (e.g., diabetic patients) and frequently carries multiple drug resistance genes, including resistance to carbapenems (2). In contrast, hvKP can infect individuals regardless of age or underlying health conditions. It is prevalent in the Asia-Pacific region, typically remains sensitive to antibiotics, but is more prone to manifest as metastatic and multi-site infections (2, 3). HvKP is also associated with severe complications, such as central nervous system infections and endophthalmitis (2, 3). *K. pneumoniae* commonly colonizes the mucosal

Address correspondence to Duoyun Li, liduoyun94@163.com, Zhijian Yu, yuzhijiansmu@163.com, or Tieying Hou, sz_houtieying@yeah.net.

Yunhui He, Fangfang Liu, and Congcong Li contributed equally to this article. The author order was determined by drawing straws.

Jiayan Wu, Kewei Fan, Zewen Wen, Duoyun Li, Zhijian Yu, and Tieying Hou contributed equally to this article.

The authors declare no conflict of interest.

surfaces of the gastrointestinal (GI) tract in both animals and healthy humans. In healthy individuals, nasopharyngeal colonization rates range from 1% to 6%, while intestinal colonization occurs in 5% to 38% of the population. Among diabetic patients, the GI colonization rate increases significantly, reaching approximately 47.5% (4, 5).

Colonization resistance (CR) refers to the ability of commensal microbiota at a given site to prevent colonization by exogenous or pathogenic bacteria (6). *K. pneumoniae* must overcome CR of the gut microbiota to colonize the GI (7). Osbelt et al. found that acid-producing *Klebsiella oxytoca* inhibited the colonization of multidrug-resistant (MDR) *K. pneumoniae* through competitive utilization of β-glucosides (including sucrose [Suc]) (8). However, this inhibition was neutralized by the addition of high concentrations of lactulose, suggesting that microbial carbohydrate metabolism plays a key role in GI colonization. Moreover, high sugar concentrations may impair CR in the gut microbiota (8, 9). Previous studies have demonstrated the pressure of carbohydrate on the metabolism, virulence, and antimicrobial susceptibility of *K. pneumoniae* (10–13). However, most studies have focused mainly on the types of carbohydrates available to gut microbes, with little attention paid to the effects of a single carbohydrate at different concentration gradients and under sustained induction. Thus, evaluating *K. pneumoniae* growth under different concentrations of sucrose and glucose (Glc) may provide theoretical insights into the mechanisms underlying CR.

Sucrose, a naturally abundant disaccharide, is found in fruits, vegetables, nuts, sugarcane, and other agricultural crops (14). It serves as a carbohydrate source for a wide range of microorganisms in natural environments, including *K. pneumoniae* and other members of the *Enterobacteriaceae* family that colonize the GI (14–16). Sucrose consists of 1 molecule of α-D-glucose and 1 molecule of β-D-fructose linked by an α-1,2 glycosidic bond, with the molecular formula $C_{12}H_{22}O_{11}$ (Fig. 1). In bacteria, sucrose is mainly transported via the phosphoenolpyruvate-dependent phosphotransferase system (PEP-PTS), which is composed of the histidine-phosphoryl protein (HPr), enzyme I (EI), and enzyme II (EII) (16, 17).

The strain NTUH-K2044 (hereinafter referred to as K2044), which is a hvKP, carries the *scrA* and *scrY* genes. *scrA* encodes the enzyme IIBC (E IIBC) of the sucrose-specific PTS system, which is a key protein in the PTS sucrose transport pathway, catalyzing the phosphorylation of sucrose and its transport across the cell membrane (12, 18). The *scrY* gene encodes a carbohydrate-specific porin, ScrY, which functions as a sucrose channel protein and enhances the sucrose transport affinity, increasing the Km by approximately 30-fold (16, 19, 20). Therefore, this study aims to reveal the effects of sustained sucrose and glucose induction on *K. pneumoniae* growth, hypermucoviscosity (HMV) phenotype, and antimicrobial susceptibility. Additionally, it examines the impact of *scrA* and *scrY* gene knockouts on bacterial growth to preliminarily elucidate the mechanisms by which sucrose and glucose influence *K. pneumoniae* growth.

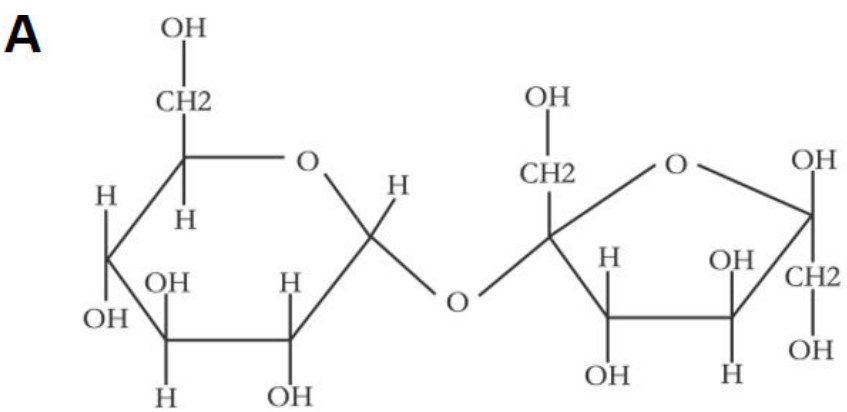

**FIG 1** Chemical structural formula for sucrose (A) and glucose (B).

## MATERIALS AND METHODS

### Bacterial strains and growth conditions

In this study, eight clinical isolates from inpatients of Shenzhen Nanshan People's Hospital, China, and one strain of NTUH-K2044 (obtained from Department of Microbiology, National Taiwan University College of Medicine, Taipei, Taiwan) were collected. All strains used in this work are listed in Table S1. *K. pneumoniae* was grown in Luria-Bertani medium at 37°C for 16–18 h, shaking with 220 rpm unless otherwise stated. For antimicrobial susceptibility testing, strains were grown in cation-adjusted Mueller-Hinton broth (CAMHB) at 37°C with shaking. *K. pneumoniae* was cultured by De Man, Rogosa, and Sharp (MRS, purchased from Topbiol, Shangdong, China, consists of peptone (10.0 g/L), beef extract (8.0 g/L), yeast extract (4.0 g/L), tween 80 (1.0g/L), sodium acetate·$H_2O$ (5.0 g/L), $K_2HPO_4$·$7H_2O$ (2.0 g/L), ammonium citrate tribasic (2.0 g/L), $MgSO_4$·$7H_2O$ (0.2 g/L), and $MnSO_4$·$4H_2O$ (0.05 g/L) for non-supplemented carbon conditions) medium in the growth curve experiments.

### Antimicrobials and reagents

Gentamicin (HY-A0276A; ≥98%), ceftriaxone sodium (HY-B0712B; 98.03%), meropenem (HY-13678; ≥98%), eravacycline (HY-16980A; 98.98%), and kanamycin (HY-16566; 99.97%) were purchased from MedChemExpress (Shanghai, China).

Sucrose and D-Glucose were purchased from Biosharp, Beijing, China. DL5000 DNA marker, bacterial DNA extraction kits, and 2× taq plus master mix II were purchased from Vanzyme, Nanjing, China. Gibson assemblycloning kit, Q5 high-fidelity DNA polymerase, and restriction enzyme StuI, SacI were obtained from New England Biolabs, America. The Gel extraction kit was purchased from Omega, America.

### Identification and passage of strains

Clinical isolates were identified by the BD Phoenix100TM system. K2044, EKP19, and EKP108 were resuscitated on LB agar and inoculated in MRS agar (supplemented with glucose or sucrose at concentrations of 0%, 0.5%, and 8%, resepectively) and cultured for 37°C, 16–24 h. Single colonies were picked and passaged in MRS broth (without sugar), as well as 0.5% sucrose, 8% sucrose, 0.5% glucose, and 8% glucose MRS broth, respectively, 37°C for 16–18 h, shaking with 220 rpm, for 60 days to obtain 3 subpassaged strains (K2044-Con-60G, EKP19-Con-60G, EKP108-Con-60G) and 12 induced strains (K2044-0.5Suc-60G, K2044-8Suc-60G, K2044-0.5Glc-60G, and K2044-8Glc-60G, same for EKP19 and EKP108), which were named in the manner of "original strain name-sugar concentration-number of generations." Strain identification was performed at 5-generation intervals during the passaging process.

### Growth curves

Overnight cultures of *K. pneumoniae* were diluted 1:500 in fresh MRS broth (without sugar), combined with 0.5%, 1%, 2%, 4%, 8%, 16% sucrose, and inoculated in 96-well polystyrene microtiter plates (200 μL/well), 37°C for 24 h, shaking type is set to intermittent, amplitude is low, and speed is set to normal, stop time before measurement, duration, and interval is 5 s, 20 s, and 20 s, respectively; MRS without sugar was used as untreated control. OD600 was determined by a Bioscreen C system (Lab Systems Helsinki Finland). The experiment was recorded for 24 h. Each assay was performed in triplicate at least three times.

### Whole-genome sequencing

Genomic DNA was extracted from K2044-8Suc-60G and K2044 with a bacterial DNA extraction kits (Vanzyme, Nanjing, China). Sequencing libraries were prepared and whole genomes were sequenced in an Illumina HiSeq2500 sequencer. Genomic alignments were performed with MUMmer4 and LASTZ 1.04.22 tools. Single-nucleotide

polymorphisms, insertions, deletions, and structural variation annotations were identified based on inter-sample alignment results with MUMmer and LASTZ.

## Construction of knockout strains

The primers used in this study are listed in Table S2. The phosphotransferase system E IIBC-encoding gene *scrA* knockout strain (K2044-Δ*scrA*) and carbohydrate porin-encoding gene *scrY* knockout strain (K2044-Δ*scrY*) of *K. pneumoniae* were constructed by homologous gene recombination technology. The vector plasmid is pPK-Ts-NgAgo-loxp-Kan. Taking *scrA* as an example, the knockout vector was first constructed. The upstream and downstream homology arms of the *scrA* gene were amplified, the pPK-Ts-NgAgo-loxp-Kan vector backbone was digested with restriction endonuclease StuI and SacI, Gibson assembly was performed, and the recombinant transformation was identified by PCR. The constructed plasmid was transformed into K2044 competent cell, recombined, and the plasmid was eliminated and picked single clones. Determine K2044-Δ*scrA* and K2044-Δ*scrY* success knockout by screening test of resistance to Kanamycin and PCR to identify *scrA* and *scrY* genes.

## String test

The strains were resuscitated and bacterial streaking on blood agar plates (Trypticase soy broth with 5% sterile defidrinated sheep blood) in a 37°C constant temperature incubator for 24 h. A sterile inoculation loop was used to pick a single colony. If the mucous thread stretched ≥5 mm, the string test was considered positive; otherwise, it was negative. The experiment was repeated three times.

## Antimicrobial susceptibility testing

Antimicrobials MICs were determined by the broth macrodilution method in cation-adjusted Mueller-Hinton broth according to Clinical and Laboratory Standards Institute guidelines (CLSI-M100-S34). The range of concentrations (twofold dilutions) tested for antimicrobials was 0.25–256 µg/mL. Antimicrobial susceptibility results were confirmed based on CLSI-M100-S34 (Table S3). All experiments were performed in triplicate.

## Graphing and statistical analysis

Data were visualized and presented as means with SDs and were analyzed in Prism 8.0 software (GraphPad Software, La Jolla, CA) and ImageJ (2.3.0). Data were processed using multiple comparisons of one-way ANOVA.

## RESULTS

### The adaptive growth of *K. pneumoniae* with sucrose pressure

To investigate the growth of *K. pneumoniae* under different concentrations of sugar stress after adaptive laboratory evolution (ALE), here in, *K. pneumoniae* K2044 was cultured *in vitro* under control conditions (without sugar), as well as in media supplemented with glucose or sucrose at concentrations of 0.5% and 8%, resepectively, and serially passaged for 60 generations. Then K2044-Con-60G was selected in control culture, while the glucose-induced strains were designated as K2044-0.5Glc-60G and K2044-8Glc-60G, and the sucrose-induced strains were labeled as K2044-0.5Suc-60G and K2044-8Suc-60G. The planktonic growth of these strains was assessed and compared in media containing a series of sucrose concentrations (0.5%, 1%, 2%, 4%, and 8%) (Fig. S1; Fig. 2). Overall, 8% or 4% of sucrose most effectively promoted the planktonic growth of K2044 and its derivatives (Fig. S1). Under sucrose exposure with the same concentration, sucrose or glucose-induced *K. pneumoniae* (K2044-8Glc-60G, K2044-0.5Suc-60G, and K2044-8Suc-60G) showed the phenotype of slowest growth, compared to K2044, K2044-Con-60G, or K2044-0.5Glc-60G, respectively

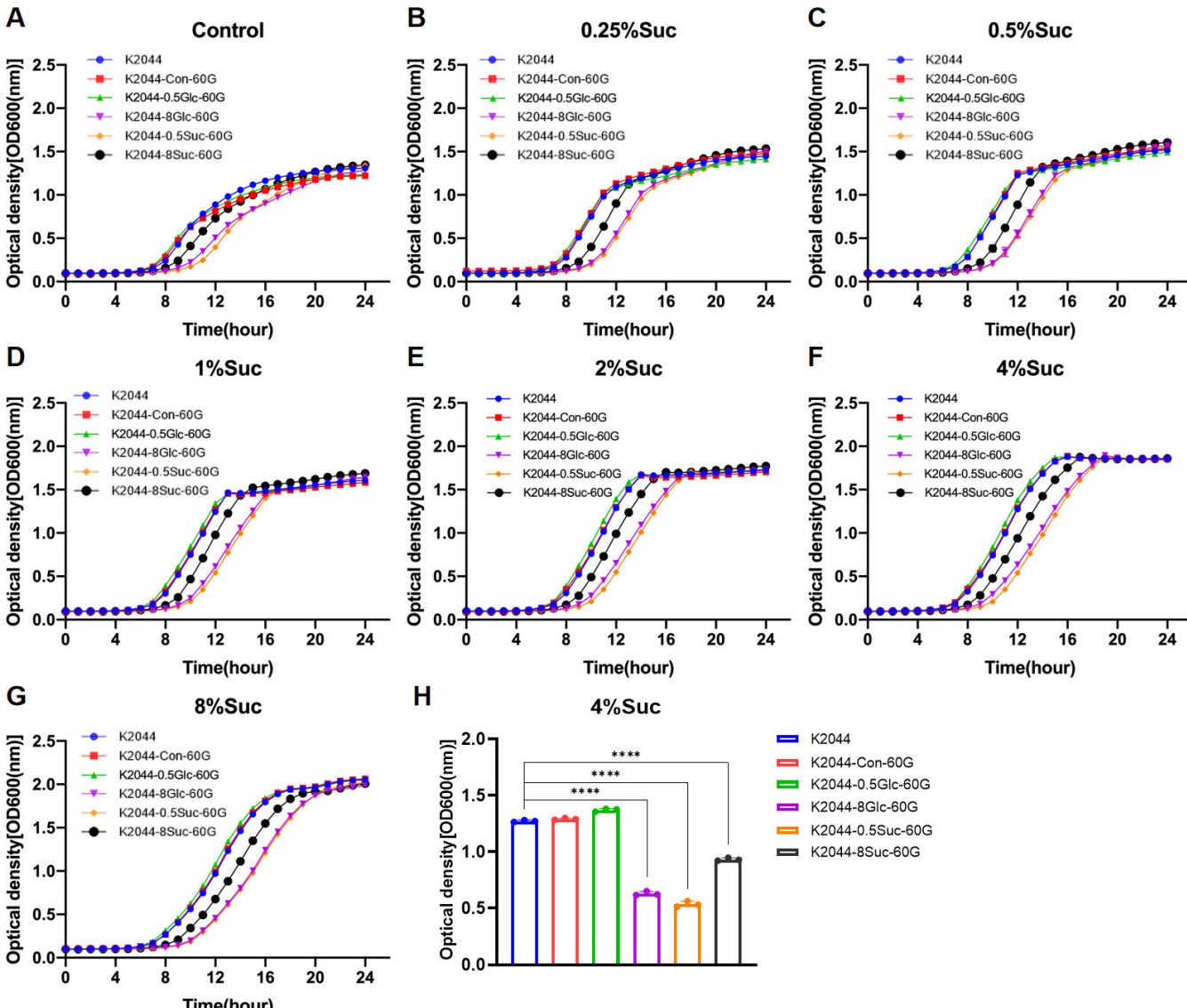

**FIG 2** Growth curves of K2044 and its derivatives (K2044-Con-60G, K2044-0.5Suc-60G, K2044-8Suc-60G, K2044-0.5Glc-60G, and K2044-8Glc-60G) at the same initial sucrose concentration. MRS without sugar as control (A), 0.25% sucrose (B), 0.5% sucrose (C), 1% sucrose (D), 2% sucrose (E), 4% sucrose (F), and 8% sucrose (G) MRS. Comparison of OD600 values of K2044 series strains at logarithmic period (12th hour), $n = 3$, ****$P < 0.0001$ (H).

(Fig. 2), suggesting that *K. pneumoniae* with the persistent induction of sucrose or glucose might not surely improve their growth capacity.

Moreover, we investigated the effects of sustained sucrose exposure on the planktonic growth of two additional *K. pneumoniae* clinical isolates, EKP19 and EKP108, along with their derived strains, respectively. EKP19-Con-60G, EKP19-0.5Glc-60G, EKP19-8Glc-60G, EKP19-0.5Suc-60G, and EKP19-8Suc-60G were selected and identified according to the same methods described for K2044. For EKP19, 4% sucrose appeared to be the optimal concentration for promoting planktonic growth (Fig. S2). Notably, EKP19-0.5Suc-60G exhibited the enhanced growth across all tested sucrose concentrations compared to the other EKP19-derived strains, especially in 4% and 8% (Fig. 3), which suggested that prior exposure to 0.5% sucrose might improve the adaptive growth of EKP19 under high-sucrose conditions. In contrast, persistent exposure to glucose or sucrose had no significant effect on the adaptive growth of EKP108 derivatives (EKP108-Con-60G, EKP108-0.5Glc-60G, EKP108-8Glc-60G, EKP108-0.5Suc-60G, and

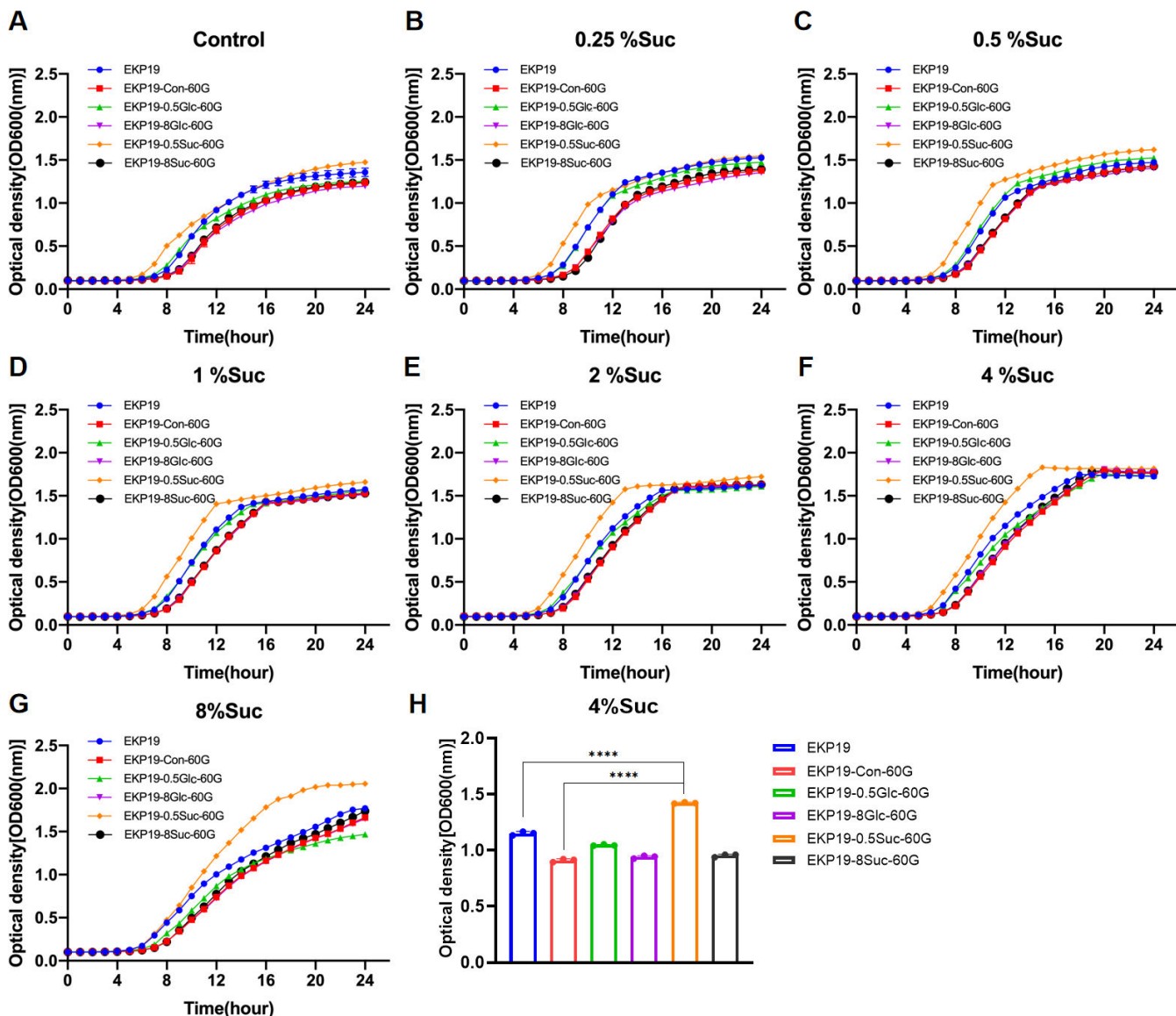

**FIG 3** Growth curves of EKP19 and its derivatives (EKP19-Con-60G, EKP19-0.5Suc-60G, EKP19-8Suc-60G, EKP19-0.5Glc-60G, and EKP19-8Glc-60G) at the same initial sucrose concentration. MRS without sugar as control (A), 0.25% sucrose (B), 0.5% sucrose (C), 1% sucrose (D), 2% sucrose (E), 4% sucrose (F), and 8% sucrose (G) MRS. Comparison of OD600 values of EKP19 series strains at logarithmic period (12th hour), $n = 3$, ****$P < 0.0001$ (H).

EKP108-8Suc-60G) (Fig. S3; Fig. 4). Additionally, the comparative growth of six other *K. pneumoniae* clinical isolates (EKP18, EKP50, EKP54, EKP72, LBKP77, LBKP79) from different clinical specimens under a series of sucrose concentration were analyzed in this study (Table S1). However, the decreased growth of these six clinical isolates at 8% sucrose was not found, which showed the different tendency compared to EKP19 (Fig. S1G through L and S2). Overall, these findings suggested that the adaptive growth response of *K. pneumoniae* to sustained glucose or sucrose exposure is strain-specific.

## The adaptive growth of *K. pneumoniae* with glucose pressure

The planktonic growth of parental isolates (K2044, EKP19, and EKP108) and their respective derivatives was evaluated under varying concentrations of glucose *in vitro*. Overall, 4% and 8% glucose appeared to be the most effective concentrations for promoting planktonic growth in these *K. pneumoniae* strains (Fig. S4 to S6). For K2044, K2044-8Glc-60G exhibited the enhanced adaptive growth compared

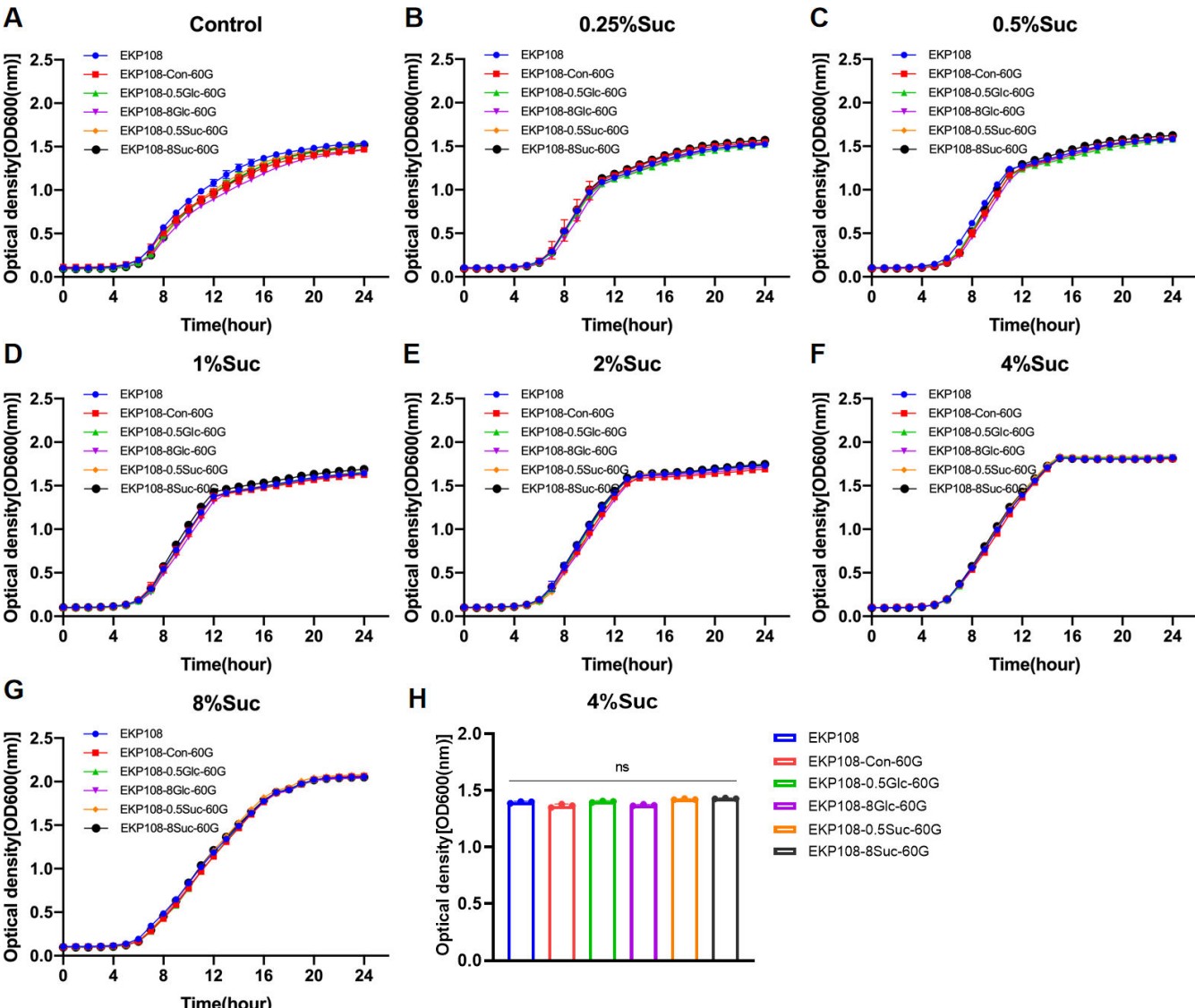

**FIG 4** Growth curves of EKP108 and its derivatives (EKP108-Con-60G, EKP108-0.5Suc-60G, EKP108-8Suc-60G, EKP108-0.5Glc-60G, and EKP108-8Glc-60G) at the same initial sucrose concentration. MRS without sugar as control (A), 0.25% sucrose (B), 0.5% sucrose (C), 1% sucrose (D), 2% sucrose (E), 4% sucrose (F), and 8% sucrose (G) MRS. Comparison of OD600 values of EKP108 series strains at logarithmic period (12th hour), $n = 3$, ns: no significance (H).

to K2044-Con-60G, K2044-0.5Glc-60G, K2044-8Suc-60G, and K2044-0.5Suc-60G under different glucose concentrations (Fig. 5). For EKP19, EKP19-0.5Suc-60G showed the improved adaptive growth compared to other EKP19-derived strains or the control when cultured in glucose-containing media (Fig. 6). In contrast, EKP108 and its derivatives (EKP108-Con-60G, EKP108-0.5Glc-60G, EKP108-8Glc-60G, EKP108-0.5Suc-60G, and EKP108-8Suc-60G) demonstrated almost unchangeable growth capacities, indicating that neither sustained glucose nor sucrose exposure significantly altered the adaptive growth of EKP108 (Fig. 7). Growth assays were also conducted on other six clinical isolates at various initial glucose concentrations. Interestingly, the growth of EKP54 seemed to be worse after the addition of serial concentrations of glucose than the control (0% glucose) (Fig. S4I).

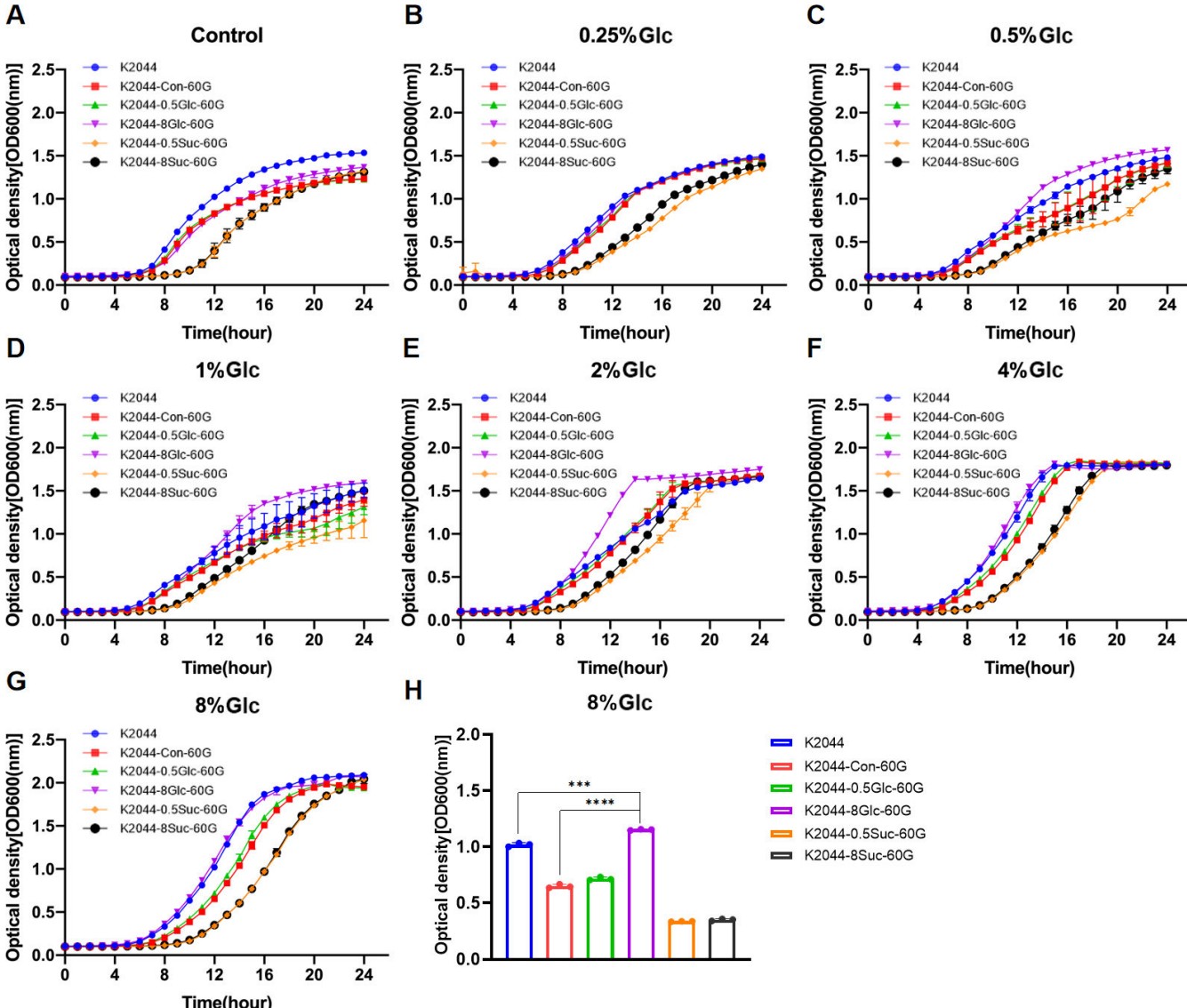

**FIG 5** Growth curves of K2044 and its derivatives (K2044-Con-60G, K2044-0.5Suc-60G, K2044-8Suc-60G, K2044-0.5Glc-60G, and K2044-8Glc-60G) at the same initial glucose concentration. MRS without sugar as control (A), 0.25% glucose (B), 0.5% glucose (C), 1% glucose (D), 2% glucose (E), 4% glucose (F), and 8% glucose (G) MRS. Comparison of OD600 values of K2044 series strains at logarithmic period (12th hour), $n = 3$, ***$P < 0.001$, ****$P < 0.0001$ (H).

## Genomic mutation of sucrose-induced *K. pneumoniae*

To further investigate the mechanisms underlying the impact of sucrose on *K. pneumoniae*, whole-genome sequencing was performed on K2044-8Suc-60G and compared to its parental strain, K2044. A total of 25 base mutation sites were identified between the two, of which 24 were located in intergenic regions and 1 was a synonymous mutation within a coding region. These data indicated that no amino acid mutation occurred in K2044 after sustained induction by 8% sucrose (K2044-8Suc-60G) (Table 1).

## Effects of sucrose and glucose on the growth of *K. pneumoniae* with the knockout of *scrA* and *scrY* genes

The *scrA* and *scrY* genes are known to play important roles in bacterial utilization of sucrose and glucose (12, 20); however, the effects of different concentrations of these sugars on the planktonic growth of *K. pneumoniae* lacking these genes should be further elucidated. Here in, we found that 4% or 8% glucose or sucrose yielded

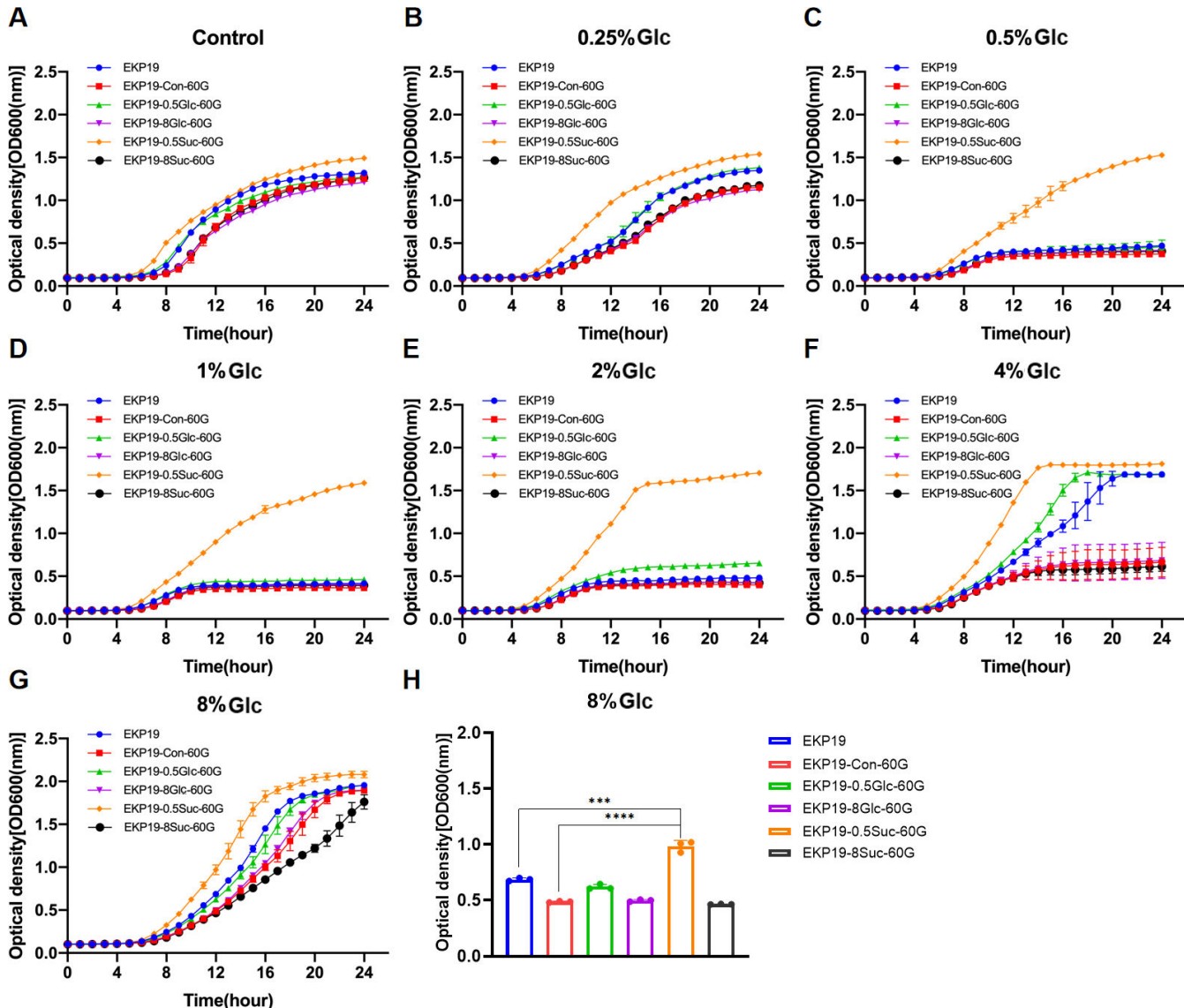

**FIG 6** Growth curves of EKP19 and its derivatives (EKP19-Con-60G, EKP19-0.5Suc-60G, EKP19-8Suc-60G, EKP19-0.5Glc-60G, and EKP19-8Glc-60G) at the same initial glucose concentration. MRS without sugar as control (A), 0.25% glucose (B), 0.5% glucose (C), 1% glucose (D), 2% glucose (E), 4% glucose (F), and 8% glucose (G) MRS. Comparison of OD600 values of EKP19 series strains at logarithmic period (12th hour), $n = 3$, ***$P < 0.001$, ****$P < 0.0001$ (H).

the enhanced planktonic growth of the *scrA* and *scrY* knockout strains (K2044-Δ*scrA* and K2044-Δ*scrY*) (Fig. S7). Nevertheless, both knockout strains exhibited significantly prolonged lag phases and reduced logarithmic-phase growth compared to the wild-type K2044 (K2044 WT) when cultured in glucose- or sucrose-containing media (Fig. 8 and 9). However, the higher the concentration of sucrose or glucose in the culture media, the more closely the growth curves of the *scrA* knockout strains resembled that of K2044 WT. These findings suggest that the deletion of *scrA* may impair the adaptive growth of *K. pneumoniae* under conditions lacking or containing low concentrations of sucrose or glucose, whereas the growth impairment resulting from *scrY* deletion cannot be compensated by increasing sugar concentrations. Notably, although both knockout strains exhibited a markedly prolonged lag phase, they ultimately reached similar peak OD values as the K2044 WT strain during the stationary phase of planktonic growth.

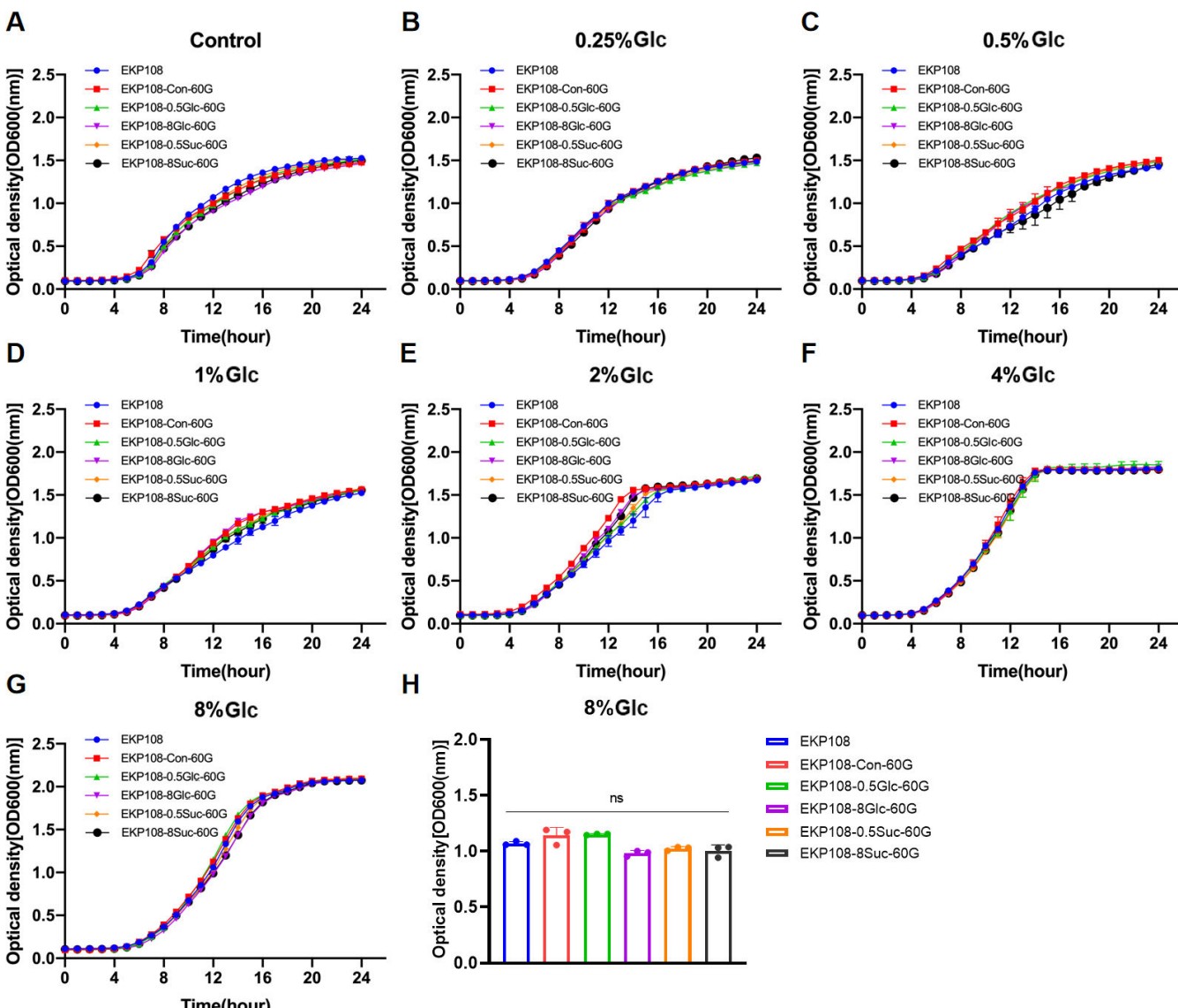

**FIG 7** Growth curves of EKP108 and its derivatives (EKP108-Con-60G, EKP108-0.5Suc-60G, EKP108-8Suc-60G, EKP108-0.5Glc-60G, and EKP108-8Glc-60G) at the same initial glucose concentration. MRS without sugar as control (A), 0.25% glucose (B), 0.5% glucose (C), 1% glucose (D), 2% glucose (E), 4% glucose (F), and 8% glucose (G) MRS. Comparison of OD600 values of EKP108 series strains at logarithmic period (12th hour), $n = 3$, ns: no significance (H).

## The mucus phenotype and antimicrobial susceptibility testing of *K. pneumoniae*

Hypermucoviscous (HMV) is one of the characteristics of highly virulent strains, and the string tests were performed on K2044 WT, K2044-Δ*scrA*, K2044-Δ*scrY*, K2044-Con-60G, K2044-0.5Glc-60G, K2044-8Glc-60G, K2044-0.5Suc-60G, and K2044-8Suc-60G to preliminarily verify whether sugar induction would have an effect on the virulence of the strain. Among these, K2044-0.5Suc-60G exhibited a significantly longer mucoviscous string compared to K2044 WT (Fig. 10), while no apparent changes in string length were observed among the other strains. These findings suggest that the deletion of the *scrA* or *scrY* genes does not significantly affect the hypermucoviscous phenotype of *K. pneumoniae*.

The antimicrobial susceptibility to gentamicin (CN), meropenem (MEM), ceftriaxone (CRO), and eravacycline (ERV) was assessed in K2044, EKP19, and EKP108 and their respective derivatives (Table 2). The antimicrobial susceptibility of CN, MEM, CRO, or ERV

**TABLE 1** Whole gene sequencing comparison between K2044-8Suc-60G and K2044

| Position | Base change | Tripletcode | Amino acids | Mutation type | Ref_gene_ID |
|---|---|---|---|---|---|
| Scaffold28_10108 | A<->G | –[a] | – | – | Intergenic region |
| Scaffold28_10111 | G<->A | – | – | – | Intergenic region |
| Scaffold28_10183 | G<->A | – | – | – | Intergenic region |
| Scaffold28_10270 | A<->G | – | – | – | Intergenic region |
| Scaffold28_10345 | G<->A | – | – | – | Intergenic region |
| Scaffold28_10348 | A<->C | – | – | – | Intergenic region |
| Scaffold28_10360 | A<->G | – | – | – | Intergenic region |
| Scaffold28_10366 | C<->T | – | – | – | Intergenic region |
| Scaffold28_10369 | C<->G | – | – | – | Intergenic region |
| Scaffold28_10489 | G<->C | – | – | – | Intergenic region |
| Scaffold28_10591 | C<->T | – | – | – | Intergenic region |
| Scaffold28_10678 | G<->C | – | – | – | Intergenic region |
| Scaffold28_9895 | C<->G | – | – | – | Intergenic region |
| Scaffold4_77398 | C<->G | – | – | – | Intergenic region |
| Scaffold5_259181 | C<->G | – | – | – | Intergenic region |
| Scaffold5_259183 | C<->G | – | – | – | Intergenic region |
| Scaffold5_259184 | A<->G | – | – | – | Intergenic region |
| Scaffold5_259188 | G<->C | – | – | – | Intergenic region |
| Scaffold5_259192 | C<->A | – | – | – | Intergenic region |
| Scaffold5_259195 | G<->C | – | – | – | Intergenic region |
| Scaffold5_259196 | C<->A | – | – | – | Intergenic region |
| Scaffold5_259197 | C<->A | – | – | – | Intergenic region |
| Scaffold5_259203 | A<->C | – | – | – | Intergenic region |
| Scaffold5_259210 | C<->A | – | – | – | Intergenic region |
| Scaffold5_266304 | C<->T | CGC<->CGT | R<->R | Synonymo-us mutation | K0_GM002169 (265279–266949 bp) |

[a]"–" indicates no change.

among K2044 WT, its derivatives, and knockout strains was compared, suggesting no significant change of resistance phenotype. Interestingly, compared to EKP19 WT, the MICs of CN and CRO in EKP19-0.5Suc-60G showed a 512-fold and 256-fold increase, respectively, while showing a significant reduction in the MIC of ERV. Additionally, compared to EKP108 WT, the MICs of CN, CRO, and ERV against EKP108-0.5Glc-60G exhibited 128-fold, 1,024-fold, and 4-fold reduction, respectively. These results suggest that sustained induction with 0.5% glucose or sucrose might randomly alter the antibiotic resistance phenotype of *K. pneumoniae*, while the deletion of the *scrA* and *scrY* genes had no significant effect on the antibiotic susceptibility.

## DISCUSSION

Glucose and sucrose are common dietary sugars and serve as primary carbon sources for most microorganisms (14). Glucose can promote the growth of *K. pneumoniae* and stimulate the production of capsular polysaccharide (CPS) and type III pili (5, 21). Sucrose facilitates biofilms formation in *Streptococcus pyogenes* and *Listeria monocytogenes* (22–24). Our results revealed that 4% or 8% glucose and sucrose are favorable for promoting adaptive growth of sugar-induced strains, consistent with previous studies (25, 26). However, the promotional effect of 8% sucrose on the adaptive growth of EKP109 series strains was reduced compared to that at 1%–4% concentrations, which may be attributed to cellular stress responses induced by hyperosmolarity and high sugar stress or excessive acid production (25, 27). Based on the theory of carbon catabolite repression (CCR), it is now generally accepted that strains preferentially consume glucose for energy before utilizing other carbon sources (28). Unexpectedly, in this study, EKP19 and EKP54 displayed reduced growth compared to controls when glucose was the sole carbon source. It is suggested that certain clinical isolates may lack the key gene

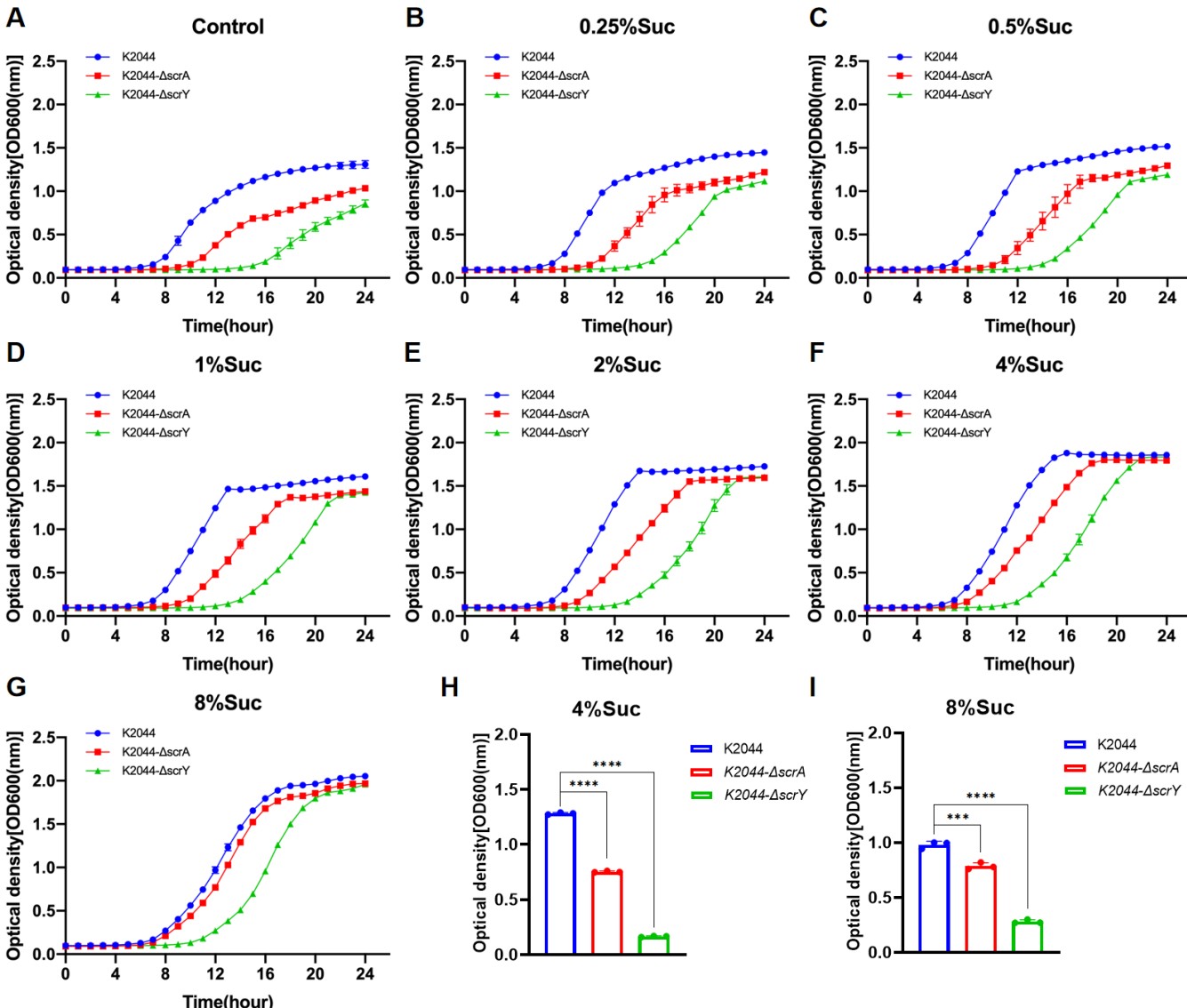

**FIG 8** Growth curves of K2044 WT, K2044-Δ*scrA*, and K2044-Δ*scrY* at the same initial sucrose concentration (A–G). Comparison of OD600 values of K2044 WT, K2044-Δ*scrA*, and K2044-Δ*scrY* at logarithmic period (12th hour), $n = 3$, ***$P < 0.001$, ****$P < 0.0001$ (H and I).

cluster involved in glucose metabolism (glycolysis-related genes), thereby limiting their ability to catabolize glucose. These findings further confirm that glucose utilization varies among *K. pneumoniae* strains.

Sustained high concentrations of glucose can promote bacterial colonization and increase the risk of infection in susceptible individuals (29). In diabetic patients, sustained exposure to elevated glucose levels increases susceptibility to various infections, including liver abscess, sepsis, endogenous endophthalmitis, and urinary tract infections (30–34). In this study, under sustained sucrose or glucose exposure, *K. pneumoniae* strains derived from K2044, EKP19, EKP108, and other clinical isolates exhibited distinct tendency of adaptive growth, indicating strain-specific differences in their responses to sugar-induced stress. The molecular mechanisms underlying these divergent adaptive phenotypes remain unclear and merit further investigation.

Whole-genome sequencing of a highly sucrose-induced strain revealed only intergenic and synonymous mutations, which demonstrated that the adaptive growth of *K. pneumoniae* under sucrose exposure might be changed without missense mutations of functional proteins. This may be attributed to growth inhibition under high sucrose

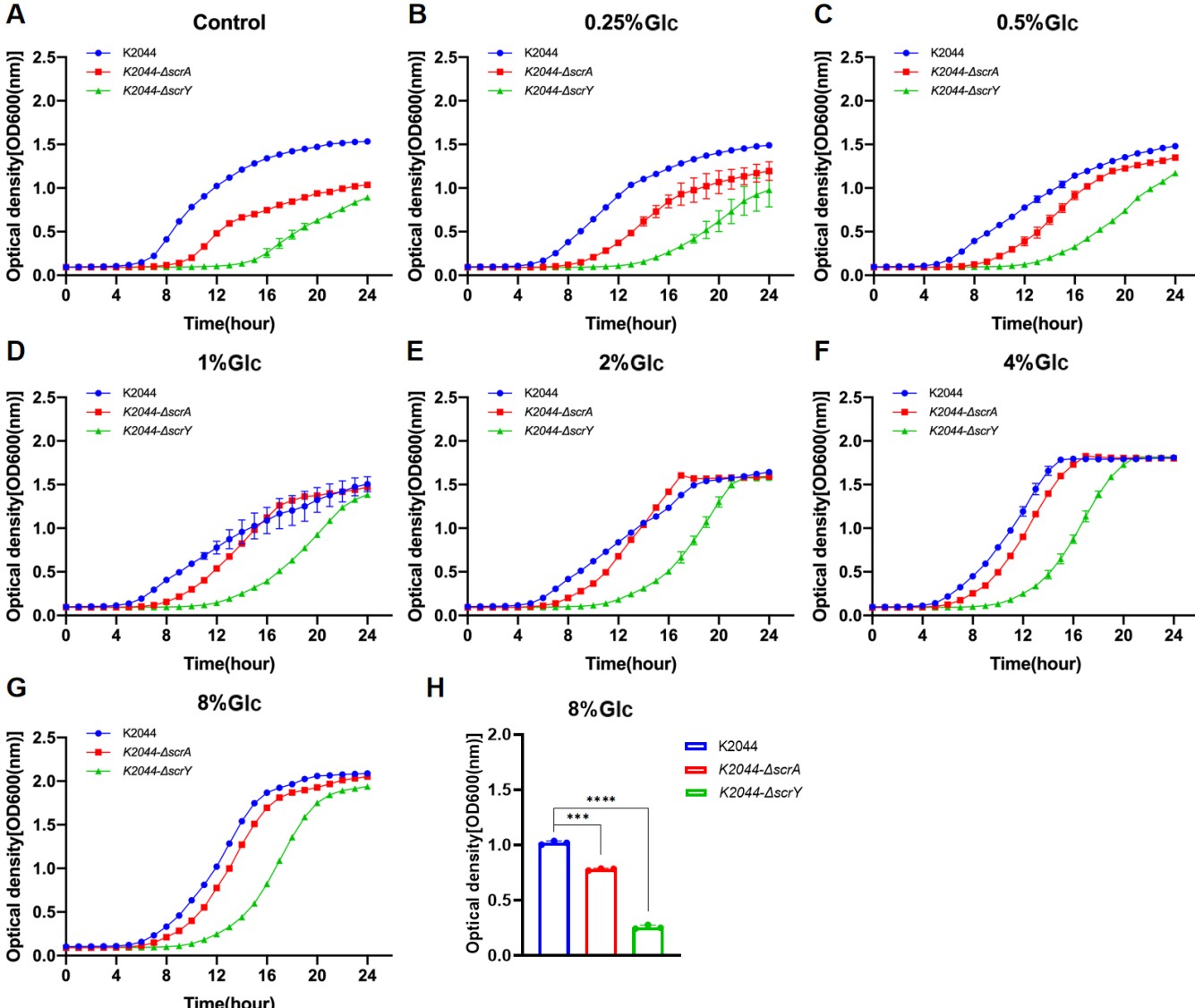

**FIG 9** Growth curves of K2044 WT, K2044-ΔscrA, and K2044-ΔscrY at the same initial glucose concentration (A–G). Comparison of OD600 values of K2044 WT, K2044-ΔscrA, and K2044-ΔscrY at logarithmic period (12th hour), $n = 3$, ***$P < 0.001$, ****$P < 0.0001$ (H).

induction, limiting the emergence of adaptive mutations. Additionally, phenotypic screening via the string test showed enhanced mucoviscosity only in certain strains after sucrose induction, suggesting improved detection of borderline HMV phenotypes. Antimicrobial resistance in *K. pneumoniae* is a growing public health concern, with resistance to MEM rising sharply in recent years (35). Although all strains in this study remained MEM-sensitive, the limited number of clinical isolates constrains broader conclusions. Notably, the high MIC values of ERV observed in some ESBL-producing strains deviate from earlier reports, highlighting potential heterogeneity in ERV resistance, which may involve tetracycline resistance genes or mutations in ribosomal subunits (36–38). Additionally, this study highlights that sustained sugar exposure can influence the antibiotic resistance phenotype of *K. pneumoniae*, and some showed unexpected changes in resistance to CN and CRO. These shifts may result from plasmid loss, mutational events, or changes in bacterial density and warrant further genomic validation.

The *scrA* gene encodes E IIBC, which plays a central role in sucrose metabolism (39). Although the genes encoding EIIBC differ among bacterial species, the deletion of the

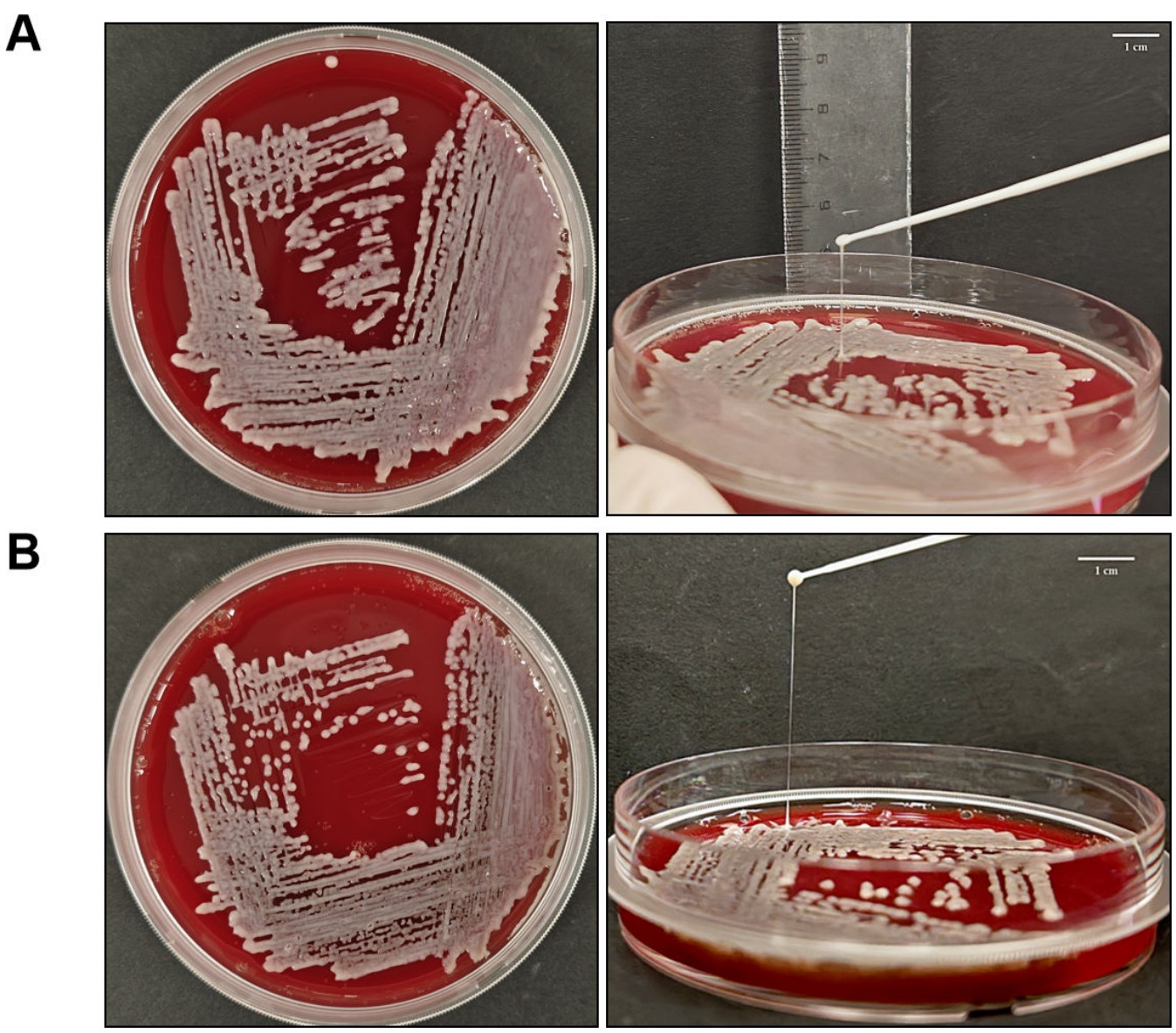

**FIG 10** Growth and string test of strain K2044 WT (A) and K2044-0.5Suc-60G (B) in blood agar. Mucoviscous string lengths of K2044 and K2044-0.5Suc-60G are 2.13 cm and 3.93 cm, respectively (≥5 mm is considered positive). Scale bar = 1 cm.

corresponding EIIBC-encoding gene consistently impairs the ability of bacteria to utilize sucrose, which can be restored upon reacquisition of the gene (40, 41). In this study, the sucrose utilization ability of K2044 was significantly decreased after knockout of the *scrA* gene. Interestingly, we observed a partial recovery of sucrose utilization in the K2044-Δ*scrA* strain at higher sucrose concentrations, suggesting that other non-PTS sucrose metabolism pathways may be activated under these conditions. Furthermore, there was a slight decrease in the ability of K2044-Δ*scrA* to utilize glucose, indicating that E IIBC may have the ability to transport or even phosphorylate glucose. The *scrY* gene encodes the carbohydrate pore protein ScrY, which functions primarily as a sucrose pore protein in *K. pneumoniae* (19, 20). In this study, the K2044-Δ*scrY* strain exhibited diminished utilization of both sucrose and glucose, suggesting that ScrY may also contribute to glucose transport in addition to its established role in sucrose uptake.

**TABLE 2** Results of *K. pneumoniae* string test and antimicrobial susceptibility testing

| Strains | String test[b] | MIC (mg/L) | | | |
|---|---|---|---|---|---|
| | | CN | MEM | CRO | ERV |
| K2044 | + | 0.25 | 0.0625 | 0.125 | 0.5 |
| K2044-Con-60G | + | 0.5 | 0.03125 | 0.125 | 0.5 |
| K2044-0.5Glc-60G | + | 0.25 | 0.03125 | 0.0625 | 0.5 |
| K2044-8Glc-60G | + | 0.25 | 0.03125 | 0.0625 | 0.25 |
| K2044-0.5Suc-60G | + | 0.25 | 0.03125 | 0.0625 | 0.5 |
| K2044-8Suc-60G | + | 0.25 | 0.03125 | 0.0625 | 0.25 |
| K2044-ΔscrA | + | 0.5 | 0.03125 | 0.0625 | 0.5 |
| K2044-ΔscrY | + | 0.25 | 0.03125 | 0.0625 | 0.5 |
| EKP19 | + | 0.25 | 0.015625 | 0.25 | 8 |
| EKP19-Con-60G | – | 0.25 | 0.015625 | 0.25 | 8 |
| EKP19-0.5Glc-60G | – | 0.25 | 0.03125 | 0.25 | 8 |
| EKP19-8Glc-60G | – | 0.25 | 0.03125 | 0.25 | 8 |
| EKP19-0.5Suc-60G | – | 128[a] | 0.03125 | >64[a] | 1 |
| EKP19-8Suc-60G | – | 0.25 | 0.03125 | 0.25 | 8 |
| EKP108 | – | 64 | 0.03125 | >64 | 2 |
| EKP108-Con-60G | – | 64 | 0.03125 | >64 | 2 |
| EKP108-0.5Glc-60G | – | 0.5[a] | 0.03125 | 0.0625[a] | 0.5[a] |
| EKP108-8Glc-60G | – | 64 | 0.03125 | >64 | 1 |
| EKP108-0.5Suc-60G | – | 128 | 0.03125 | >64 | 4 |
| EKP108-8Suc-60G | – | 128 | 0.0625 | >64 | 4 |
| ATCC 25922 | N | 1 | 0.03125 | 0.0625 | 0.0625 |

[a]Changes in strain resistance phenotypes, compared to their respective wild-type strains. The antimicrobial susceptibility breakpoints are shown in Table S3. N, No experiments conducted or no results.
[b]"+" indicates positive and "−" indicates negative.

## Conclusions

In summary, this study demonstrated that 4% or 8% glucose and sucrose are favorable for promoting adaptive growth of sugar-induced *K. pneumoniae* strains. Strain-specific differences were observed in glucose utilization, indicating heterogeneity in carbohydrate metabolism among *K. pneumoniae* clinical isolates. In addition, most sustained sugar induction had no significant effect on the hypermucoviscous phenotype of *K. pneumoniae* as assessed by the string test, but it may influence antibiotic susceptibility in certain strains. The deletion of *scrA* may impair the adaptive growth of *K. pneumoniae* under conditions lacking or containing low concentrations of sucrose or glucose, whereas the growth impairment resulting from *scrY* deletion cannot be compensated by increasing sugar concentrations. However, the absence of either gene did not significantly alter antimicrobial susceptibility or the hypermucoviscous phenotype of *K. pneumoniae*. Collectively, this work elucidates the impact of single-carbon source gradients and sustained sugar induction on the adaptive growth and drug resistance of *K. pneumoniae* and preliminarily reveals the roles of *scrA* and *scrY* in carbohydrate metabolism, suggesting a possible mechanism by which sucrose and glucose affect the adaptive growth of *K. pneumoniae*. These findings contribute to the theoretical understanding of CCR and provide insights that may inform clinical management of *K. pneumoniae*-related infections.

## ACKNOWLEDGMENTS

This work was supported by the following grants: National Natural Science Foundation of China (82172283, 82572621); Guangdong Basic and Applied Basic Research Foundation (2024A1515013276); Sanming Project of Medicine in Shenzhen (SMZ202303037); Shenzhen Key Medical Discipline Construction Fund (SZXK06162); Science, Technology and Innovation Commission of Shenzhen Municipality of basic research funds (JCYJ20240813114518024, KJZD20240903103500002) and the Shenzhen Nanshan

District Scientific Research Program of the People's Republic of China (NSZD2025005, NSZD2024038, NS2023049, NS2024001, NSZD2023019, NSZD2024023, NS2024007Y, NS2024044Y).

## AUTHOR AFFILIATIONS

[1]School of Basic Medical Sciences, Shenzhen University Medical School, Shenzhen University, Shenzhen, China
[2]Department of Infectious Diseases and the Key Lab of Endogenous Infection, Affiliated Nanshan Hospital of Shenzhen University, Shenzhen, China
[3]School of Pharmacy, Shenzhen University Medical School, Shenzhen University, Shenzhen, China

## AUTHOR ORCIDs

Duoyun Li 🆔 http://orcid.org/0009-0004-4313-4487
Zhijian Yu 🆔 http://orcid.org/0000-0002-5677-2064
Tieying Hou 🆔 http://orcid.org/0000-0002-3902-2632

## AUTHOR CONTRIBUTIONS

Yunhui He, Data curation, Formal analysis, Investigation, Methodology, Software, Visualization, Writing – original draft, Writing – review and editing | Fangfang Liu, Data curation, Formal analysis, Investigation, Methodology, Software, Visualization, Writing – review and editing | Congcong Li, Data curation, Formal analysis, Investigation, Methodology, Software, Writing – review and editing | Jiayan Wu, Formal analysis, Methodology, Validation, Writing – review and editing | Kewei Fan, Formal analysis, Methodology, Validation, Writing – review and editing | Zewen Wen, Methodology, Project administration, Supervision, Writing – review and editing | Duoyun Li, Funding acquisition, Project administration, Resources, Supervision, Writing – review and editing | Zhijian Yu, Conceptualization, Funding acquisition, Project administration, Resources, Supervision, Writing – review and editing | Tieying Hou, Funding acquisition, Project administration, Resources, Supervision, Writing – review and editing

## DATA AVAILABILITY

The raw whole-genome sequencing data wasdata were posted in the Sequence Read Archive (SRA) database under accession number SRR19634253 (K2044-8Suc-60G) and SRR23562543 (K2044 WT) (http://www.ncbi.nlm.nih.gov/sra).

## ETHICS APPROVAL

All methods were carried out in accordance with relevant guidelines and regulations and were approved by the Ethics Committee of Shenzhen Nanshan People's Hospital and the 1964 Helsinki declaration and its later amendments, or comparable ethical standards. The biosafety approval number is 0125F300106. All experimental procedures involving human subjects were approved by the institutional ethical committee of Shenzhen Nanshan People's Hospital. Bacterial strains were obtained from stored samples of hospitalized patients, collected as part of the routine clinical management of patients, according to the national guidelines in China. Therefore, informed consent was not sought, and informed consent waiver was approved by the institutional ethical committee of Shenzhen Nanshan People's Hospital.

## ADDITIONAL FILES

The following material is available online.

## Supplemental Material

**SNP results S1 (Spectrum01603-25-s0001.xls).** SNP results.
**SNP results S2 (Spectrum01603-25-s0002.xls).** SNP results.
**SNP results S3 (Spectrum01603-25-s0003.xls).** SNP results.
**SNP results S4 (Spectrum01603-25-s0004.xls).** SNP results.
**SNP results S5 (Spectrum01603-25-s0005.txt).** SNP results.
**SNP results S6 (Spectrum01603-25-s0006.txt).** SNP results.
**SNP results S7 (Spectrum01603-25-s0007.txt).** SNP results.
**SNP results S8 (Spectrum01603-25-s0008.txt).** SNP results.
**SNP results S9 (Spectrum01603-25-s0009.txt).** SNP results.
**Supplemental tables and figures (Spectrum01603-25-s0010.docx).** Tables S1 to S3 and Figures S1 to S7.

## Open Peer Review

**PEER REVIEW HISTORY (review-history.pdf).** An accounting of the reviewer comments and feedback.

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
