## [Reviewer comments · Microbiology Spectrum]

Microbiology Spectrum

The Adaptive Growth and Mechanisms of *Klebsiella pneumoniae* under Sucrose and Glucose Exposure.

Yunhui HE, Fangfang LIU, Congcong LI, Jiayan WU, Kewei FAN, Zewen Wen, Duoyun Li, Zhijian Yu, and Tieying Hou

Corresponding Author(s): Zhijian Yu, Shenzhen University Medical School

Review Timeline:

Submission Date:	May 23, 2025
Editorial Decision:	July 26, 2025
Revision Received:	August 22, 2025
Editorial Decision:	September 14, 2025
Revision Received:	September 22, 2025
Accepted:	September 25, 2025

Editor: Deena Altman

Reviewer(s): The reviewers have opted to remain anonymous.

Transaction Report:

DOI: <https://doi.org/10.1128/spectrum.01603-25>

Re: Spectrum01603-25 (**The Adaptive Growth and Mechanisms of *Klebsiella pneumoniae* under Sucrose and Glucose Exposure.**)

Dear Prof. Zhijian Yu:

Thank you for the privilege of reviewing your work. Below you will find my comments, instructions from the Spectrum editorial office, and the reviewer comments.

Revision Guidelines

Sincerely,
Deena Altman
Editor
Microbiology Spectrum

Reviewer #1 (Comments for the Author):

Summary

This study was performed based on the premise that *K. pneumoniae* must compete for sugars to overcome colonization resistance, particularly sucrose. Thus the authors evolved a hvKp strain (K2044) and multiple ESBL clinical isolates in different

concentrations of glucose and sucrose for 60 generations. The authors then performed growth assays to assess changes in bacterial sugar utilization, string tests to test changes in mucoidy, and MIC breakpoints to examine changes in antibiotic resistance. A variety of phenotypes was reported. Since the lineages had individually evolved it is likely that the authors are presenting possible (stochastic) evolutionary outcomes and conclusions should be carefully drawn. While the data generally seem robust, more quantitative and statistical analyses of the data should be performed to support the authors' conclusions.

Major critiques

1. Line 90-96: The methods for passaging strains are sparse and unclear. Are they passaged in liquid medium, plated on LB agar (with or without sugar?) and then re-inoculated in liquid medium. What are the 3 sub-passaged strains vs 12 induced strains (line 94-95)? Perhaps a schematic would be helpful.
2. Most conclusions are drawn about single isolates from long evolution experiments, so claims about cause and effect (e.g. 0.5% suc and 8% glc do not enhance growth under sucrose) should not be made. More independently evolved replicates would be needed to make such broad claims. Likely what outcomes are observed for individual lineages is stochastic. This critique could be addressed by editing the language of the conclusions.
3. Each figure should be accompanied by relevant statistics used in the figure. Many conclusions are not supported by statistical analyses of the presented data. While the data may appear to support the conclusion, quantitative, statistical analyses of the existing data would greatly strengthen the manuscript. The use of terminologies such as 'best growth' when defining bacterial growth is arbitrary. Even if the arbitrary terms were to be used, this should be followed by the discussion of quantifiable observations such as cumulative growth (cumulative OD), final culture density, growth rate, duration of lag and doubling time. Extracting these quantifiable growth properties from the growth curves could greatly strengthen this manuscript. This suggestion applies to discussion of all growth curves throughout the text. Some specific examples are identified below:
 - a. Lines 142-143 - Since K2044-8Glu-60G, K2044-0.5Suc-60G, and K2044-8Suc-60G all have significantly lower OD than K2044 (I believe) in Fig 2H it would be more accurate to conclude that all three evolved lineages do not elicit improved growth in sucrose.
 - b. Lines 152-154 and 176-178 and 179-180 and 180-183 - The conclusions are not supported by statistical analyses of the data as presented. Mid-log OD₆₀₀ data, analogous to Fig 2H and 3H, could be used to justify the conclusions. It might make sense to use 8% Glc (C₆H₁₂O₆) data to approximate the amount of carbon available in 4% Suc (C₁₂H₂₂O₁₁) condition.
 - c. Line 207-211 and 212-213: The conclusions are not supported by statistical analyses of the data as presented. This could be remedied by performing statistical analyses of OD₆₀₀ at 12 hr as in Fig 2 and 3. For lines 212-213 a correlation analysis of mid-log OD vs Suc or Glc concentration could justify the conclusions.
 - d. Line 224-225: The rationale for selecting HMV change in response to glc/suc is unclear, especially if most strains are classical lineage. No control is shown and no quantification (data) are provided. Without a visual control and/or quantification of the data, the conclusions are not justified. Performing a sedimentation assay will likely provide more robust data for making claims about changes in mucoidy.
4. It seems that the authors have used the 27th edition of CLSI-M100 for reference. There are several new editions published by CLSI after the 27th edition. It is critical to follow up-to-date standards for the methods (MIC) and breakpoints used to classify antimicrobial sensitivity. At least, it should be ensured by cross-reference that the breakpoints between the 27th edition and the recent edition are similar for the antibiotics used in this study.

Minor critiques

5. Methods: All methods are missing important information. Specific examples are identified below:
 - a. The method used to knock out scrA and scrY are neither cited nor discussed in detail.
 - b. Describe what MRS medium is or provide a reference.
 - c. Line 78: Information about duration of culture and shaking parameter (rpm) is missing.
 - d. Line 97-102: Please specify incubation temperature and shaking parameter.
 - e. Line 106: What versions of MUMer and LASTZ were used?
 - f. Line 117: How was the blood agar prepared? Was it a commercially available or prepared in-house?
6. For bacterial growth curves, logarithmic scale is recommended to plot OD₆₀₀. Log scale makes the data points for early log/exponential phase visible.
7. References missing for: Line 37-38, 46-48, 205-207
8. Is the K2044 strain actually NTUH-K2044? If so, at minimum, please use the full name in the methods/strains table. Are the clinical isolates classical or hypervirulent pathotypes?
9. Strain table should have reference to the literature from which strains were sourced. It seems like the Reference column is a

mixture of isolation source and reference. These pieces of information (clinical isolation site vs strain source) should be distinct and captured in separate columns.

10. The standard abbreviation for glucose is Glc, not Glu (which could be confused with glutamate).

11. How many replicates are shown in the growth curves? What statistical tests were applied? These should be reported in the figure legends.

12. Is Fig S1A-E replotted data from Fig 2? If so, that should be explicitly stated in the figure legend.

13. Line 152-153 - Do you mean "...had no significant effect on the adaptive growth of EKP108 derivatives"? By definition EKP108 parent was not persistently exposed to sugars.

14. Fig S5 - The different media conditions seem to produce very erratic results. How many times was the media prepared and/or experiment run? Is it possible the media was not prepared properly or not inoculated well? It is very strange that in S5C there is growth in 1 and 2% Glc, but not for the other panels. 4% Glc is rather variable too. This variability may merit repeating the data to verify reproducibility of the reported phenotypes. Most of the other growth curves in this manuscript appear comparable between panels, thus raising concerns about experimental issues confounding Fig S5 only.

15. Line 196 - Typo: surose

16. Table 1: The nucleotide location of the SNPs should be reported using published NTUH-K2044 genome base numbering.

17. Line 198: It appears that the lab stock of parental K2044 was not sequenced. It could be possible that some of the variations mentioned in Table 1 could have arisen from the lab stock of parental K2044 itself. If, WGS is not performed then this limitation should be included in the results and/or discussion.

18. Line 229-231: The sentence should be edited to make it clear whether all strains or just K2044, EKP19 and EKP108 were included in antimicrobial susceptibility testing.

19. Line 232 - Typo: derivatives

20. Table 2: Asterisk is not defined. Assuming it indicates statistical significance, a statement about the statistical tests used should be reported. Abbreviations should be defined: CN, MEM, CRO, ERV, and N (next to ATCC 25922). Also is MIC in mg/L?

21. Table 2: There are instances of slight changes in MICs of ERV and CN for K2044 adapted strains. This is not discussed or acknowledged in the results or discussion. Are these differences significant in terms of CLSI breakpoint antimicrobial sensitivity?

22. Line 253-254: What does "key gene cluster" mean in this case? Is it in reference to glycolytic genes or transport-associated genes?

23. Line 319: If available, biosafety approval # should be mentioned for transparency.

Reviewer #2 (Comments for the Author):

I appreciate that He et al in this article have done a lot of CCR-related work, which is not easy to demonstrate due to complexity of the network. This manuscript is very hard to read with the current writing and presentation especially for the results section; discussion is still fine. Manuscript will benefit from some clear context and rationale to each experimental question postulated, followed by presentation of only the most relevant data, and then drawing a concise conclusion. Further, in the current version of the manuscript there were so many instances where there is no finite conclusion.

1. This manuscript will benefit from the model figure for each of the figures presented. All the growth data should go into supplemental except for the most vital to each figure.

2. In introduction author needs to explain why cKP and hvKP are different genetically...what makes them different, if not known then mention that.

3. Line 64 of the provide context for the strain K2044 as to what is it..(it is described in the method section but would be good to know while reading the intro)

4. Line 132 should read "in-vitro"

5. Define MRS media, and enlist its composition for non-supplemented carbon conditions

6. Why did you not use any sort of a defined minimal media for KP in their study?

7. In the results section for each experimental data or procedure rationale needs to be provided currently it is missing for every single data and experiments for example, (line 131-143). Without that how reader will know why you did the experiment...

8. In a ctrl strain without sucrose what is the source of carbon that allows for growth till OD600nm =1? How much sugar is in the

MRS?

9. Fig 1 panel H needs to show all the data points that were used to generate the bar graph. As per the method $n=3$ then all the data points should be shown. Also, what statistical test was used, what groups were compared, and was there any post-correction test applied for multiple comparisons?

10. "A total of 25 base mutation sites were identified between the two, of which 24 were located in intergenic regions and one was a synonymous mutation within a coding region." But where in the reference genome position or between ORFs it is present? Authors need to provide a clear information.

11. Additionally, there is already a high baseline difference among K2044-0.5Suc-60G vs EKP19-0.5Suc-60G for the growth on 4% sucrose, would it be worth to sequences these strains instead of just the K2044 with K2044-8Glu-60G. What allows EKP19 to better manage 0.5Suc than the K2044?

12. Did you try RT-PCR to see the transcription of HPr and EI in the K2044 with K2044-8Glu-60G? As I believe some of the endogenous machinery is shared by the PTS system so perhaps high sugar somehow affects EI and HPr activity or transcription?

Summary

This study was performed based on the premise that *K. pneumoniae* must compete for sugars to overcome colonization resistance, particularly sucrose. Thus the authors evolved a hvKp strain (K2044) and multiple ESBL clinical isolates in different concentrations of glucose and sucrose for 60 generations. The authors then performed growth assays to assess changes in bacterial sugar utilization, string tests to test changes in mucoidy, and MIC breakpoints to examine changes in antibiotic resistance. A variety of phenotypes was reported. Since the lineages had individually evolved it is likely that the authors are presenting possible (stochastic) evolutionary outcomes and conclusions should be carefully drawn. While the data generally seem robust, more quantitative and statistical analyses of the data should be performed to support the authors' conclusions.

Major critiques

1. Line 90-96: The methods for passaging strains are sparse and unclear. Are they passaged in liquid medium, plated on LB agar (with or without sugar?) and then re-inoculated in liquid medium. What are the 3 sub-passaged strains vs 12 induced strains (line 94-95)? Perhaps a schematic would be helpful.
2. Most conclusions are drawn about single isolates from long evolution experiments, so claims about cause and effect (e.g. 0.5% suc and 8% glc do not enhance growth under sucrose) should not be made. More independently evolved replicates would be needed to make such broad claims. Likely what outcomes are observed for individual lineages is stochastic. This critique could be addressed by editing the language of the conclusions.
3. Each figure should be accompanied by relevant statistics used in the figure. Many conclusions are not supported by statistical analyses of the presented data. While the data may appear to support the conclusion, quantitative, statistical analyses of the existing data would greatly strengthen the manuscript. The use of terminologies such as 'best growth' when defining bacterial growth is arbitrary. Even if the arbitrary terms were to be used, this should be followed by the discussion of quantifiable observations such as cumulative growth (cumulative OD), final culture density, growth rate, duration of lag and doubling time. Extracting these quantifiable growth properties from the growth curves could greatly strengthen this manuscript. This suggestion applies to discussion of all growth curves throughout the text. Some specific examples are identified below:
 - a. Lines 142-143 – Since K2044-8Glu-60G, K2044-0.5Suc-60G, and K2044-8Suc-60G all have significantly lower OD than K2044 (I believe) in Fig 2H it would be more accurate to conclude that all three evolved lineages do not elicit improved growth in sucrose.

- b. Lines 152-154 and 176-178 and 179-180 and 180-183 – The conclusions are not supported by statistical analyses of the data as presented. Mid-log OD600 data, analogous to Fig 2H and 3H, could be used to justify the conclusions. It might make sense to use 8% Glc (C₆H₁₂O₆) data to approximate the amount of carbon available in 4% Suc (C₁₂H₂₂O₁₁) condition.
 - c. Line 207-211 and 212-213: The conclusions are not supported by statistical analyses of the data as presented. This could be remedied by performing statistical analyses of OD600 at 12 hr as in Fig 2 and 3. For lines 212-213 a correlation analysis of mid-log OD vs Suc or Glc concentration could justify the conclusions.
 - d. Line 224-225: The rationale for selecting HMV change in response to glc/suc is unclear, especially if most strains are classical lineage. No control is shown and no quantification (data) are provided. Without a visual control and/or quantification of the data, the conclusions are not justified. Performing a sedimentation assay will likely provide more robust data for making claims about changes in mucoidy.
4. It seems that the authors have used the 27th edition of CLSI-M100 for reference. There are several new editions published by CLSI after the 27th edition. It is critical to follow up-to-date standards for the methods (MIC) and breakpoints used to classify antimicrobial sensitivity. At least, it should be ensured by cross-reference that the breakpoints between the 27th edition and the recent edition are similar for the antibiotics used in this study.

Minor critiques

5. Methods: All methods are missing important information. Specific examples are identified below:
- a. The method used to knock out scrA and scrY are neither cited nor discussed in detail.
 - b. Describe what MRS medium is or provide a reference.
 - c. Line 78: Information about duration of culture and shaking parameter (rpm) is missing.
 - d. Line 97-102: Please specify incubation temperature and shaking parameter.
 - e. Line 106: What versions of MUMer and LASTZ were used?
 - f. Line 117: How was the blood agar prepared? Was it a commercially available or prepared in-house?
6. For bacterial growth curves, logarithmic scale is recommended to plot OD600. Log scale makes the data points for early log/exponential phase visible.

7. References missing for: Line 37-38, 46-48, 205-207
8. Is the K2044 strain actually NTUH-K2044? If so, at minimum, please use the full name in the methods/strains table. Are the clinical isolates classical or hypervirulent pathotypes?
9. Strain table should have reference to the literature from which strains were sourced. It seems like the Reference column is a mixture of isolation source and reference. These pieces of information (clinical isolation site vs strain source) should be distinct and captured in separate columns.
10. The standard abbreviation for glucose is Glc, not Glu (which could be confused with glutamate).
11. How many replicates are shown in the growth curves? What statistical tests were applied? These should be reported in the figure legends.
12. Is Fig S1A-E replotted data from Fig 2? If so, that should be explicitly stated in the figure legend.
13. Line 152-153 – Do you mean “...had no significant effect on the adaptive growth of EKP108 derivatives”? By definition EKP108 parent was not persistently exposed to sugars.
14. Fig S5 – The different media conditions seem to produce very erratic results. How many times was the media prepared and/or experiment run? Is it possible the media was not prepared properly or not inoculated well? It is very strange that in S5C there is growth in 1 and 2% Glc, but not for the other panels. 4% Glc is rather variable too. This variability may merit repeating the data to verify reproducibility of the reported phenotypes. Most of the other growth curves in this manuscript appear comparable between panels, thus raising concerns about experimental issues confounding Fig S5 only.
15. Line 196 – Typo: surose
16. Table 1: The nucleotide location of the SNPs should be reported using published NTUH-K2044 genome base numbering.
17. Line 198: It appears that the lab stock of parental K2044 was not sequenced. It could be possible that some of the variations mentioned in Table 1 could have arisen from the lab stock of parental K2044 itself. If, WGS is not performed then this limitation should be included in the results and/or discussion.
18. Line 229-231: The sentence should be edited to make it clear whether all strains or just K2044, EKP19 and EKP108 were included in antimicrobial susceptibility testing.
19. Line 232 – Typo: derivatives
20. Table 2: Asterisk is not defined. Assuming it indicates statistical significance, a statement about the statistical tests used should be reported. Abbreviations should be defined: CN, MEM, CRO, ERV, and N (next to ATCC 25922). Also is MIC in mg/L?

21. Table 2: There are instances of slight changes in MICs of ERV and CN for K2044 adapted strains. This is not discussed or acknowledged in the results or discussion. Are these differences significant in terms of CLSI breakpoint antimicrobial sensitivity?
22. Line 253-254: What does “key gene cluster” mean in this case? Is it in reference to glycolytic genes or transport-associated genes?
23. Line 319: If available, biosafety approval # should be mentioned for transparency.

Dear editor:

We have revised the paper according to the reviewers' suggestions and highlighted the revisions in yellow in the revised manuscript.

Moreover, we have given a point-to-point response to the reviewers' questions and showed them as below. Meanwhile, we have carefully checked the words and grammar in the article to ensure that there are no errors.

Thank you for any response and please contact us if you or the reviewers have any questions about our revised manuscript.

Sincerely,

Tieying HOU MD

Zhijian YU MD

Department of Infectious Diseases and Shenzhen key lab for endogenous infection, Shenzhen Nanshan people's Hospital, Shenzhen University. No 89, Taoyuan Road, Nanshan district, Shenzhen 518052, China

Email: yuzhijiansmu@163.com

Reviewer #1:

Summary

This study was performed based on the premise that *K. pneumoniae* must compete for sugars to overcome colonization resistance, particularly sucrose. Thus the authors evolved a hvKp strain (K2044) and multiple ESBL clinical isolates in different concentrations of glucose and sucrose for 60 generations. The authors then performed growth assays to assess changes in bacterial sugar utilization, string tests to test changes in mucoidy, and MIC breakpoints to examine changes in antibiotic resistance. A variety of phenotypes was reported. Since the lineages had individually evolved it is likely that the authors are presenting possible (stochastic) evolutionary outcomes and conclusions should be carefully drawn. While the data generally seem robust, more quantitative and statistical analyses of the data should be performed to support the authors' conclusions.

Major critiques

1. Line 90-96: The methods for passaging strains are sparse and unclear. Are they passaged in liquid medium, plated on LB agar (with or without sugar?) and then re-inoculated in liquid medium. What are the 3 sub-passaged strains vs 12 induced strains (line 94-95)? Perhaps a schematic would be helpful.

Response: Thank you for your suggestion and we have added more detailed information in the revised manuscript. You can see it in MATERIALS AND METHODS (Line 117-125).

2. Most conclusions are drawn about single isolates from long evolution experiments, so claims about cause and effect (e.g. 0.5% suc and 8% glc do not enhance growth under sucrose) should not be made. More independently evolved replicates would be needed to make such broad claims. Likely what outcomes are observed for individual lineages is stochastic. This critique could be addressed by editing

the language of the conclusions.

Response: Thank you for your suggestion .In this study, we just aimed to elucidate the isolate-specific characteristics of *K.pneumoniae* under glucose or sucrose pressure. Therefore, we make the conclusion according to the results using the specific *K.pneumoniae* isolates. we have revised our description in our revised manuscript.

For example, the following was shown the modified description in the first paragraph of RESULTS (Line 185-189):

Original version: Under sucrose exposure with the same concentration, K2044-0.5Glu-60G exhibited the best growth, whereas K2044-8Glu-60G and K2044-0.5Suc-60G showed the phenotype of slowest growth (Figure 2), suggesting that sustained sucrose pressure with 0.5% and glucose pressure with 8% might not enhance the growth capacity of K2044 under sucrose exposure.

Revised version: Under sucrose exposure with the same concentration, sucrose or glucose-induced *K.pneumoniae* (K2044-8Glu-60G, K2044-0.5Suc-60G and K2044-8Suc-60G) showed the phenotype of slowest growth, compared to K2044, K2044-Con-60G or K2044-0.5Glu-60G, respectively (Figure 2), suggesting that *K.pneumoniae* with the persistent induction of sucrose or glucose might not surely improve their growth capacity.

3. Each figure should be accompanied by relevant statistics used in the figure. Many conclusions are not supported by statistical analyses of the presented data. While the data may appear to support the conclusion, quantitative, statistical analyses of the existing data would greatly strengthen the manuscript. The use of terminologies such as 'best growth' when defining bacterial growth is arbitrary. Even if the arbitrary terms were to be used, this should be followed by the discussion of quantifiable observations such as cumulative growth (cumulative OD), final culture density, growth rate, duration of lag and doubling time. Extracting these quantifiable growth properties from the growth curves could greatly strengthen this manuscript. This suggestion applies to discussion of all growth curves throughout the text. Some specific examples are identified below:

Response: Thank you for your suggestion.In the revised manuscript, we have increased the statistical analyses of the presented data according to the demand of our conclusions. The arbitrary terminologies in the article regarding “best growth” etc. have been changed. We chose the OD600 values of planktonic bacteria to quantify the growth of the strains according to our previous report and other references.The OD600 values during the logarithmic period (12th hour in this study) were also used to compare the growth of planktonic bacteria because during the logarithmic phase, the increase in OD is positively correlated with the increase in the number of viable bacteria. In fact, our unreported data supported the consistency of the OD600 values and CFU counts during the logarithmic period, therefore we think the reliability of this method has been demonstrated.

Main references:

- 1) DOI: 10.1038/s41564-024-01886-5, in extended data.
- 2) DOI: 10.1186/s12866-025-03961-1
- 3) DOI: 10.1080/22221751.2024.2321981.

a. Lines 142-143 - Since K2044-8Glu-60G, K2044-0.5Suc-60G, and K2044-8Suc-60G all have significantly lower OD than K2044 (I believe) in Fig 2H it would be more accurate to conclude that all three evolved lineages do not elicit improved growth in sucrose.

Response: Thank you for your suggestions. This has been modified in line 185-189 of the revised

manuscript.

b. Lines 152-154 and 176-178 and 179-180 and 180-183 - The conclusions are not supported by statistical analyses of the data as presented. Mid-log OD600 data, analogous to Fig 2H and 3H, could be used to justify the conclusions. It might make sense to use 8% Glc (C₆H₁₂O₆) data to approximate the amount of carbon available in 4% Suc (C₁₂H₂₂O₁₁) condition.

Response: Thank you for your advice. We have supplemented the statistical analysis of the OD600 values for the logarithmic period to each of the growth curve plots. Whereas, it's not our aim to demonstrate whether 8% Glc (C₆H₁₂O₆) data was approximately equal to the amount of carbon available in 4% Suc (C₁₂H₂₂O₁₁) condition.

c. Line 207-211 and 212-213: The conclusions are not supported by statistical analyses of the data as presented. This could be remedied by performing statistical analyses of OD600 at 12 hr as in Fig 2 and 3. For lines 212-213 a correlation analysis of mid-log OD vs Suc or Glc concentration could justify the conclusions.

Response: Thank you for your suggestion. We have added the statistical analysis of OD600 at 12 hr to each of the growth curve plots involving comparisons.

d. Line 224-225: The rationale for selecting HMV change in response to glc/suc is unclear, especially if most strains are classical lineage. No control is shown and no quantification (data) are provided. Without a visual control and/or quantification of the data, the conclusions are not justified. Performing a sedimentation assay will likely provide more robust data for making claims about changes in mucoidy.

Response: Thank you for your suggestion. In this study, we mainly focus on the adaptive growth of *K.pneumoniae* under glucose or sucrose pressure. Therefore, we just evaluate the impact of glucose or sucrose pressure on mucoidy changes of *K.pneumoniae* by string test to preliminarily verify the relationship between sugar induction and virulence.

The results of the string test for K2044 and K2044-0.5Suc-60G have been added to the revised manuscript in more detail. Of course, the further virulence comparisons should be performed in the future with multiple methods, including a sedimentation assay or other methods.

4. It seems that the authors have used the 27th edition of CLSI-M100 for reference. There are several new editions published by CLSI after the 27th edition. It is critical to follow up-to-date standards for the methods (MIC) and breakpoints used to classify antimicrobial sensitivity. At least, it should be ensured by cross-reference that the breakpoints between the 27th edition and the recent edition are similar for the antibiotics used in this study.

Response: Thank you for your suggestion. We have revised it to version CLSI-M100-34. In fact, the breakpoints and methods involved in this study have no difference between CLSI-M100-27 and CLSI-M100-34.

Minor critiques

5. Methods: All methods are missing important information. Specific examples are identified below:

a. The method used to knock out scrA and scrY are neither cited nor discussed in detail.

Response: Thank you for your suggestion. We have added more details in line 144-154 of the revised manuscript.

b. Describe what MRS medium is or provide a reference.

Response: Thank you for your suggestion. We have described in “Bacterial strains and growth conditions” of MATERIALS AND METHODS (Line 100-104).

De Man, Rogosa and Sharp (MRS, purchased from Topbiol, Shangdong, China, consists of peptone, beef extract, yeast extract, tween 80, sodium acetate • H₂O, K₂HPO₄ • 7H₂O, ammonium citrate tribasic, MgSO₄ • 7H₂O and MnSO₄ • 4H₂O for non-supplemented carbon conditions)

c. Line 78: Information about duration of culture and shaking parameter (rpm) is missing.

Response: Thank you for your suggestion. We have added in line 98-99, shaking parameter is 220rpm.

d. Line 97-102: Please specify incubation temperature and shaking parameter.

Response: Thank you for your suggestion. We have added in line 130-131. 37°C for 24h, shaking type is set to intermittent, amplitude is low, and speed is set to normal, stop time before measurement, duration, and interval is 5s, 20s and 20s, respectively.

e. Line 106: What versions of MUMer and LASTZ were used?

Response: Thank you for your suggestion. We have added in line 139-140. Genomic alignments were performed with MUMmer4 and LASTZ 1.04.22 tools.

f. Line 117: How was the blood agar prepared? Was it a commercially available or prepared in-house?

Response: Thank you for your suggestion. We have added in “String test” of Methods (Line 157-158). It’s prepared in-house. TSB+5% Sterile Defibrinated Sheep Blood.

6. For bacterial growth curves, logarithmic scale is recommended to plot OD₆₀₀. Log scale makes the data points for early log/exponential phase visible.

Response: Thank you for your professional advice. According to our report and others’ studies (DOI: 10.1186/s12866-025-03961-1, DOI: 10.1038/s41564-024-01886-5), the OD₆₀₀ values can be used as the representative of the bacteria growth and shows good consistency with the logarithmic scale. Therefore, we believe that the OD₆₀₀ values representing the *K.pneumoniae* growth will not affect the results of this study. In fact, our previous report showed that in the early logarithmic/exponential phase, the OD₆₀₀ values of *K.pneumoniae* was consistent with the logarithmic CFU (data not shown).

7. References missing for: Line 37-38, 46-48, 205-207

Response: Thank you for your suggestion. We have added in line 53-54, 62-64, 276-277.

8. Is the K2044 strain actually NTUH-K2044? If so, at minimum, please use the full name in the methods/strains table. Are the clinical isolates classical or hypervirulent pathotypes?

Response: Thank you for your suggestion. K2044 is actually NTUH-K2044 and we have corrected this mistake in the revised article. In this study, K2044 was determined to be hypervirulent. For clinical isolates, EKP19 is hypervirulent (string test: +, *rmpA*: +), the other clinical strains were not tested for virulence but were supplemented with sequence typing (ST) in Table S1.

9. Strain table should have reference to the literature from which strains were sourced. It seems like the Reference column is a mixture of isolation source and reference. These pieces of information (clinical isolation site vs strain source) should be distinct and captured in separate columns.

Response: Thank you for your suggestion. We have accurately summarized the origin of all strains in this study.

10. The standard abbreviation for glucose is Glc, not Glu (which could be confused with glutamate).

Response: Thank you for your professional advice. According to the articles previously reported, either Glu or Glc can be used as abbreviated forms of glucose (e.g., the following articles all use Glu for glucose: DOI: 10.1152/ajpendo.00127.2022; DOI:10.1080/17461391.2017.1317035 ; DOI: 10.1016/j.jhazmat.2022.130421; DOI: 10.1007/s00125-016-4197-8).

11. How many replicates are shown in the growth curves? What statistical tests were applied? These should be reported in the figure legends.

Response: Thank you for your suggestion. In the methods, it is mentioned that each assay was performed in triplicate at least three times (reported in the figure legends). Data were processed using statistical comparisons with one-way ANOVA (Line 171-172).

12. Is Fig S1A-E replotted data from Fig 2? If so, that should be explicitly stated in the figure legend.

Response: Thank you for your suggestion. we have restated it in the Figure S1 legend of revised manuscript.

13. Line 152-153 - Do you mean "...had no significant effect on the adaptive growth of EKP108 derivatives"? By definition EKP108 parent was not persistently exposed to sugars.

Response: Thank you for your suggestion. This sentence has been corrected to "In contrast, persistent exposure to glucose or sucrose had no significant effect on the adaptive growth of EKP108 derivatives" in line 198-199 of revised manuscript.

14. Fig S5 - The different media conditions seem to produce very erratic results. How many times was the media prepared and/or experiment run? Is it possible the media was not prepared properly or not inoculated well? It is very strange that in S5C there is growth in 1 and 2% Glc, but not for the other panels. 4% Glc is rather variable too. This variability may merit repeating the data to verify reproducibility of the reported phenotypes. Most of the other growth curves in this manuscript appear comparable between panels, thus raising concerns about experimental issues confounding Fig S5 only.

Response: Thank you for your suggestions. Each experiment was repeated three times. Combined with the previous results, the induced strain EKP19-0.5Suc-60G consistently showed better adaptive growth, so it may not be surprising that it was the only one that grew well under 1 and 2% glucose stress. As for 4% glucose, three replicates did show that EKP19-0.5Glu-60G and EKP19-0.5Suc-60G grew better than the other derivatives at 4% glucose stress. We were also surprised by this result, which may be exactly where the value of our study lies. However, the specific causes and mechanisms still need to be further explored.

15. Line 196 - Typo: surose

Response: Thank you for your suggestion. We have made the correction in line 265.

16. Table 1: The nucleotide location of the SNPs should be reported using published NTUH-K2044 genome base numbering.

Response: Thank you for your suggestion. We have added in Table 1. Only important information has been retained in the table, for more information, it is recommended to go to our uploaded sequencing results on NCBI. Accession number are SRR19634253 (K2044-8Suc-60G) and SRR23562543 (K2044 WT) (<http://www.ncbi.nlm.nih.gov/sra>).

17. Line 198: It appears that the lab stock of parental K2044 was not sequenced. It could be possible that some of the variations mentioned in Table 1 could have arisen from the lab stock of parental K2044 itself. If, WGS is not performed then this limitation should be included in the results and/or discussion.

Response: Thank you for your professional advice. In fact, we have sequenced the whole genome of the parental strain K2044 as well, and we have added the sequence number of K2044 to the "DATA AVAILABILITY STATEMENT" (line 440-441).

18. Line 229-231: The sentence should be edited to make it clear whether all strains or just K2044, EKP19 and EKP108 were included in antimicrobial susceptibility testing.

Response: Thank you for your suggestion. We have made the correction in line 315-317.

19. Line 232 - Typo: derivatives

Response: Thank you for your suggestion. We have made the correction in line 317.

20. Table 2: Asterisk is not defined. Assuming it indicates statistical significance, a statement about the statistical tests used should be reported. Abbreviations should be defined: CN, MEM, CRO, ERV, and N (next to ATCC 25922). Also is MIC in mg/L?

Response: Thank you for your suggestion. The asterisk and "N" have been defined in the revised manuscript. The full name and abbreviations of the antibiotics are stated in the article in that paragraph. The CLSI specifies that the MIC can be expressed in either $\mu\text{g/mL}$ or mg/L .

21. Table 2: There are instances of slight changes in MICs of ERV and CN for K2044 adapted strains. This is not discussed or acknowledged in the results or discussion. Are these differences significant in terms of CLSI breakpoint antimicrobial sensitivity?

Response: Thank you for your suggestion. In terms of CLSI breakpoints antimicrobial sensitivity, these minor changes did not involve alteration of the drug-sensitive phenotype. All our experiments were repeated three times, and the MICs of the quality control strain was within the normal range. These minor changes could also be caused by serial induction or knockout, which were not discussed in detail because the drug-sensitive phenotype was unchanged.

22. Line 253-254: What does "key gene cluster" mean in this case? Is it in reference to glycolytic genes or transport-associated genes?

Response: The key gene cluster refers mainly to manipulators consisting of glycolysis-related genes

rather than transport-related genes. The *glpFK* manipulator, for example, is functionally regulated by the glucose-CcpA axis. This has been added in line 372.

23. Line 319: If available, biosafety approval # should be mentioned for transparency.

Response: Thank you for your suggestion. We have added this proof No. in line 447.

Reviewer #2 (Comments for the Author):

I appreciate that He et al in this article have done a lot of CCR-related work, which is not easy to demonstrate due to complexity of the network. This manuscript is very hard to read with the current writing and presentation especially for the results section; discussion is still fine. Manuscript will benefit from some clear context and rationale to each experimental questions postulated, followed by presentation of only the most relevant data, and then drawing a concise conclusion. Further, in the current version of the manuscript there were so many instances where there is no finite conclusions.

1. This manuscript will benefit from the model figure for each of the figures presented. All the growth data should go into supplemental except for the most vital to each figure.

Response: Thank you for your advice. In this study, we aimed to investigate the isolate-specific characteristic of *K.pneumoniae* under sucrose or glucose pressure, so the growth of each strain in the presence of sugars at each concentration gradient has been retained for better comparison. The figures concerning three isolates were chosen to make readers compare their characteristics easily and rapidly.

2. In introduction author needs to explain why cKP and hvKP are different genetically...what makes them different, if not known then mention that.

Response: Thank you for your advice. We have added the description in line 46-50.

3. Line 64 of the provide context for the strain K2044 as to what is it..(it is described in the method section but would be good to know while reading the intro)

Response: Thank you for your advice. We have added the description in line 80.

4. Line 132 should read "in-vitro"

Response: Thank you for your advice. We have corrected it in line 177.

5. Define MRS media, and enlist its composition for non-supplemented carbon conditions

Response: Thank you for your advice. We have added this description in “Bacterial strains and growth conditions” (line 100-104).

De Man, Rogosa and Sharp (MRS, purchased from Topbiol, Shangdong, China, consists of peptone, beef extract, yeast extract, tween 80, sodium acetate • H₂O, K₂HPO₄ • 7H₂O, ammonium citrate tribasic, MgSO₄ • 7H₂O and MnSO₄ • 4H₂O for non-supplemented carbon conditions)

6. Why did you not use any sort of a defined minimal media for KP in their study?

Response: Thank you for your advice. We have also conducted experiments with LB broth previously.

However, since we needed to use sugar-free medium as a control for this study, we found that even sugar-free LB medium still had potential sugar present, whereas sugar-free MRS medium is more dependent by sugar (no glucose, etc.), which interfered less with this study (data not shown). In fact, It's also the cause that we used MRS media in our previous study [DOI: 10.1186/s12866-025-03961-1].

7. In the results section for each experimental data or procedure rationale needs to be provided currently it is missing for every single data and experiments for example, (line 131-143). Without that how reader will know why you did the experiment...

Response: Thank you for your advice. A rationale has been added before the results in the revised manuscript

8. In a ctrl strain without sucrose what is the source of carbon that allows for growth till OD600nm =1? How much sugar is in the MRS?

Response: Thank you for your advice. For non-supplemented carbon conditions, MRS consists of peptone, beef extract, yeast extract, tween 80, sodium acetate •H₂O, K₂HPO₄ •7H₂O, ammonium citrate tribasic, MgSO₄ •7H₂O and MnSO₄ •4H₂O. *K.pneumoniae* might utilize components from peptones and yeast extracts as alternative carbon sources for facilitating its growth. In fact, it's difficult to make clear the types and contents of carbohydrates in the sugar-free MRS or other media. In this study ,we just aimed to evaluate the isolate-specific characteristics of *K.pneumoniae* uder glucose or sucrose pressure. Therefore, we used the sugar-free media as the control in every step and our data could support the conclusion. Totally to say, our conclusions and results were reliable and dependent by this strategy.

9. Fig 1 panel H needs to show all the data points that were used to generate the bar graph. As per the method n=3 then all the data points should be shown. Also, what statistical test was used, what groups were compared, and was there any post-correction test applied for multiple comparisons?

Response: Thank you for your advice. The statistical test used has been supplemented in the “Graphing and statistical analysis” of Methods (line 171-172). As per the method n=3, all the data points have been shown.

10. "A total of 25 base mutation sites were identified between the two, of which 24 were located in intergenic regions and one was a synonymous mutation within a coding region." But where in the reference genome position or between ORFs it is present? Authors need to provide a clear information.

Response: Thank you for your advice. We provided detailed mutation site in “Position” of Table 1. Only important information has been retained in the Table 1, for more information, it is recommended to go to our uploaded sequencing results on NCBI. Accession number are SRR19634253 (K2044-8Suc-60G) and SRR23562543 (K2044 WT) (<http://www.ncbi.nlm.nih.gov/sra>).

11. Additionally, there is already a high baseline difference among K2044-0.5Suc-60G vs EKP19-0.5Suc-60G for the growth on 4% sucrose, would it be worth to sequences these strains instead of just the K2044 with K2044-8Glu-60G. What allows EKP19 to better manage 0.5Suc than the K2044?

Response: Thank you for your advice. In this study, we just aimed to investigate the adaptive growth and mechanisms of *K.pneumoniae* under sucrose and glucose exposure. By the whole-genome comparison between K2044 and K2044-8Suc-60G, we didn't found the amino acid mutations of the

functional proteins in K2044-8Suc-60G and this data has demonstrated that the adaptive growth of *K.pneumoniae* under sucrose exposure might be changed without missense mutations of functional proteins (this explanation has also been included in the DISCUSSION section, line 376-378). Moreover, It's difficult to conclude that EKP19 is better to manage 0.5Suc than the K2044. We think the data just support the isolate-specific adaption of *K.pneumoniae* under sucrose and glucose exposure. *K.pneumoniae* might exhibit better growth adaption in the persistent of low concentration of sucrose or glucose. However, different *K.pneumoniae* strains might have different adaption ability according to the same sugar pressure.

12. Did you try RT-PCR to see the transcription of HPr and EI in the K2044 with K2044-8Glu-60G? As I believe some of the endogenous machinery is shared by the PTS system so perhaps high sugar somehow affects EI and HPr activity or transcription?

Response: Thank you for your advice. The aim of this study was to investigate the effects of sustained sugar induction on the adaptive growth of *Klebsiella pneumoniae*, for which a more reliable phenotype has been obtained. Whole genome sequencing was also performed to analyze the cause of differential growth. In addition, the growth of K2044 under various concentrations of sugar stress was indeed affected after we knocked out sucrose-specific PTS system-related genes (*scrA*, *scrY*). The next step will observe the transcription of HPr and EI in the K2044 with K2044-8Glu-60G, so as to reveal the mechanism of this differential growth more deeply.

Re: Spectrum01603-25R1 (**The Adaptive Growth and Mechanisms of *Klebsiella pneumoniae* under Sucrose and Glucose Exposure.**)

Dear Prof. Zhijian Yu:

Thank you for the privilege of reviewing your work. Below you will find my comments, instructions from the Spectrum editorial office, and the reviewer comments.

Revision Guidelines

Sincerely,
Deena Altman
Editor
Microbiology Spectrum

Reviewer #1 (Public repository details (Required)):

Genomic data needs to be deposited. SRA numbers are listed.

Reviewer #1 (Comments for the Author):

Summary

The authors have considered all prior suggestions and addressed most concerns. Below are a few points that may have been missed or were not adequately addressed.

Minor critiques

5b. Describe what MRS medium is or provide a reference.

Reference or concentrations of the individual components are missing.

6. For bacterial growth curves, logarithmic scale is recommended to plot OD600. Log scale makes the data points for early log/exponential phase visible.

The authors misunderstood my suggestion. I did not mean to include CFU data on log scale. I meant that plotting OD600 values on log scale could improve visualization of early log/exponential phase.

8. Is the K2044 strain actually NTUH-K2044? If so, at minimum, please use the full name in the methods/strains table.

The use of K2044 as an abbreviation for NTUH-K2044 is never explicitly stated.

10. The standard abbreviation for glucose is Glc, not Glu (which could be confused with glutamate).

Glc is the internationally recognized abbreviation for Glucose, not Glu. Because some publications use Glu does not mean that it is standard. Internationally agreed upon standards for sugar descriptions are detailed here:

<https://www.ncbi.nlm.nih.gov/glycans/snfg.html> and PMID: 31184695.

20. Table 2: Asterisk is not defined. The authors state "**: Changes in strain resistance phenotypes", but what parameters (e.g. statistics) were used to identify changes in resistance are not provided.

Dear editor:

We thank you and the reviewers for the kind consideration and constructive comments on our manuscript.

We have revised the manuscript again based on the reviewers' suggestions, with all modifications highlighted in yellow in the revised version. Moreover, we have given a point-to-point response to the reviewers' questions and showed them as below. Meanwhile, we have carefully checked the words and grammar in the article to ensure that there are no errors, and hope that you will find the added information suitable and sufficient for publication.

Thank you for any response and please contact us if you or the reviewers have any questions about our revised manuscript.

Sincerely,

Tieying HOU MD

Zhijian YU MD

Department of Infectious Diseases and Shenzhen key lab for endogenous infection, Shenzhen Nanshan people's Hospital, Shenzhen University. No 89, Taoyuan Road, Nanshan district, Shenzhen 518052, China

Email: yuzhijiansmu@163.com

Reviewer #1 (Public repository details (Required)):

Genomic data needs to be deposited. SRA numbers are listed.

Response: Thank you. We have also uploaded the raw data for this section to the journal's submission website. The raw whole-genome sequencing data was deposited in the Sequence Read Archive (SRA) database under accession number SRR19634253 (K2044-8Suc-60G) and SRR23562543 (K2044 WT) (<http://www.ncbi.nlm.nih.gov/sra>) (Line 440-442).

Reviewer #1 (Comments for the Author):

Summary

The authors have considered all prior suggestions and addressed most concerns. Below are a few points that may have been missed or were not adequately addressed.

Minor critiques

5b. Describe what MRS medium is or provide a reference.

Reference or concentrations of the individual components are missing.

Response: Thank you for your suggestion. In fact, we have successfully used this system of MRS medium with or without xylose for monitoring the impact of xylose on the *K.pneumonia* growth in our previous report [**BMC Microbiology, DOI:10.1186/s12866-025-03961-1**], which is also our reference.

In this revised manuscript, we have added concentrations of the individual components in "Bacterial strains and growth conditions" of MATERIALS AND METHODS (Line 100-104). De Man, Rogosa and Sharp medium (MRS), purchased from Topbiol, Shangdong, China, consists

of peptone (10.0g/L), beef extract (8.0g/L), yeast extract (4.0g/L), tween 80 (1.0g/L), sodium acetate • H₂O (5.0g/L), K₂HPO₄ • 7H₂O (2.0g/L), ammonium citrate tribasic (2.0g/L), MgSO₄ • 7H₂O (0.2g/L) and MnSO₄ • 4H₂O (0.05g/L) for non-supplemented carbon conditions.

6. For bacterial growth curves, logarithmic scale is recommended to plot OD₆₀₀. Log scale makes the data points for early log/exponential phase visible.

The authors misunderstood my suggestion. I did not mean to include CFU data on log scale. I meant that plotting OD₆₀₀ values on log scale could improve visualization of early log/exponential phase.

Response: Thank you for your suggestion. In fact, OD₆₀₀ value is the original data from the automatic growth curve instrument and can directly reflect the dynamics of bacterial growth. We use the OD values directly according many previous reports and we list some here as below: DOI:10.1021/acsinfecdis.4c00293;

- DOI: 10.1038/s41538-022-00171-1;
- DOI: 10.1080/22221751.2024.2321981;
- DOI: 10.1186/s12866-025-03961-1;
- DOI: 10.1038/s42003-022-03899-4.

Plotting OD₆₀₀ values on log scale could indeed improve visualization of early log/exponential phase. However, the direct OD₆₀₀ values in this study can show more details of data changes during the bacterial growth during the lag phase, logarithmic phase, and stationary phase. The OD₆₀₀ values representing the *K.pneumoniae* growth will not affect the results of this study. Therefore, OD₆₀₀ values in the bacterial growth curve is also a visual choice in the revised manuscript. Of course, if you or reviewers stick to the transformation from OD value to logarithmic scale, please tell us, we can also revise it. Thank your suggestion again.

8. Is the K2044 strain actually NTUH-K2044? If so, at minimum, please use the full name in the methods/strains table.

The use of K2044 as an abbreviation for NTUH-K2044 is never explicitly stated.

Response: Thank you for your suggestion. We have clearly stated this in the revised manuscript (Line 80).

10. The standard abbreviation for glucose is Glc, not Glu (which could be confused with glutamate).

Glc is the internationally recognized abbreviation for Glucose, not Glu. Because some publications use Glu does not mean that it is standard. Internationally agreed upon standards for sugar descriptions are detailed here: <https://www.ncbi.nlm.nih.gov/glycans/snfg.html> and PMID: 31184695.

Response: Thank you for your professional and rigorous advice. We have amended all instances of the abbreviation “glucose” to “Glc” throughout the article.

20. Table 2: Asterisk is not defined. The authors state “*: Changes in strain resistance phenotypes”, but what parameters (e.g. statistics) were used to identify changes in resistance are not provided.

Response: Thank you for your suggestion. Changes in resistance phenotypes were confirmed based on CLSI-M100-S34, and we have listed the related information of antimicrobial

susceptibility breakpoints in TABLE S3. We have also provided explanations in the methods section and in the asterisk (Line 166 and 357-358).

Re: Spectrum01603-25R2 (**The Adaptive Growth and Mechanisms of *Klebsiella pneumoniae* under Sucrose and Glucose Exposure.**)

Dear Prof. Zhijian Yu:

Your manuscript has been accepted, and I am forwarding it to the ASM production staff for publication. Your paper will first be checked to make sure all elements meet the technical requirements. ASM staff will contact you if anything needs to be revised before copyediting and production can begin. Otherwise, you will be notified when your proofs are ready to be viewed.

Sincerely,
Deena Altman
Editor
Microbiology Spectrum